# THINK BEFORE YOU DIFFUSE: INFUSING PHYSICAL RULES INTO VIDEO DIFFUSION

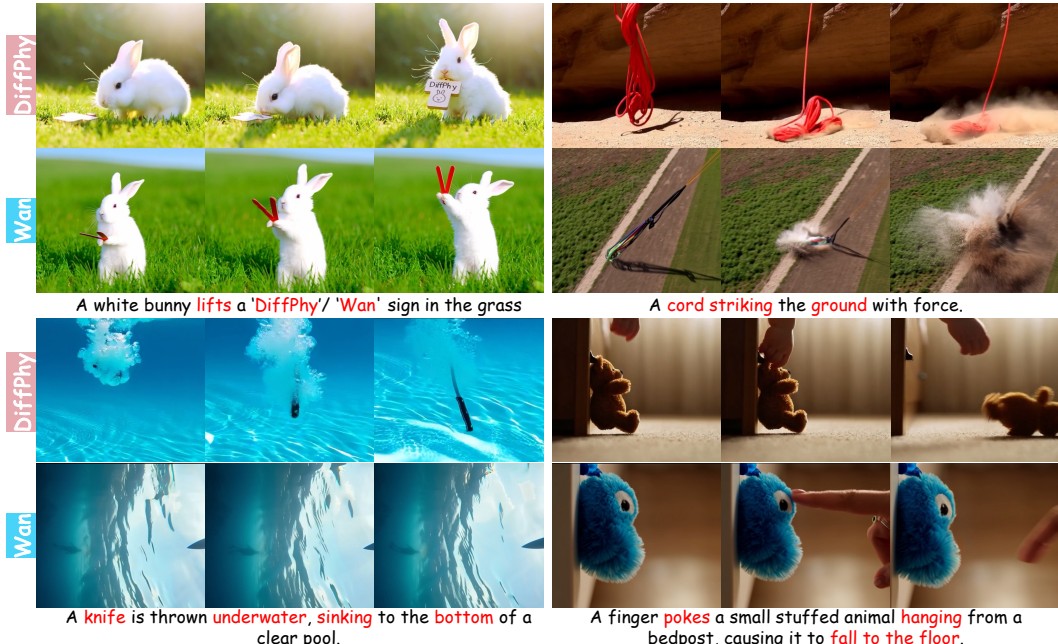

Figure 1: DiffPhy enables physically grounded and semantically aligned video generation across diverse real-world scenarios, including gravity-driven motion, fluid interactions, forceful impacts, and object manipulation. It outperforms state-of-the-art video diffusion model Wan 2.1–14B Wan (2025) in both visual plausibility and physical coherence.

## ABSTRACT

Recent video diffusion models have demonstrated their great capability in generating visually-pleasing results, while synthesizing the correct physical effects in generated videos remains challenging. The complexity of real-world motions, interactions, and dynamics introduce great difficulties when learning physics from data. In this work, we propose DiffPhy, a generic framework that enables physically-correct and photo-realistic video generation by fine-tuning a pre-trained video diffusion model. Our method leverages large language models (LLMs) to infer rich physical context from the text prompt. To incorporate this context into the video diffusion model, we use a multimodal large language model (MLLM) to verify intermediate latent variables against the inferred physical rules, guiding the model's gradient updates accordingly. MLLM's textual output is transformed into continuous signals. We then formulate a set of training objectives that jointly ensure physical accuracy and semantic alignment with the input text. Additionally, failure facts of physical phenomena are corrected via attention injection. We also establish a high-quality physical video dataset containing diverse phyiscal actions and events to facilitate effective finetuning. Extensive experiments on public benchmarks demonstrate that DiffPhy is able to produce state-of-the-art results across diverse physics-related scenarios. Code and data will be made available post-review.

## 1    INTRODUCTION

Recent advances in video diffusion models have made it possible to generate high-fidelity videos with rich details from text prompts Guo et al. (2024); Wan (2025); Yang et al. (2024b); Chen et al. (2024); Kong et al. (2024); Agarwal et al. (2025); OpenAI (2024); LumaAI (2025); Wang et al. (2023); Fan et al. (2025); Pika; Runway (2024); KlingAI (2024); Mei & Patel (2023). Such generation capability has opened up a wide range of real-world applications, including movie production Zhang et al. (2025a), gaming Valevski et al. (2024), and even serving as virtual environments for robotics and embodied AI learning Agarwal et al. (2025); Alhaija et al. (2025). Despite the great visual quality, accurately simulating the correct physical-related effects remains challenging and is often overlooked by existing methods Kang et al. (2024); Xue et al. (2024). These models learn physics by exhaustively training the model over a large corpus of video data. The complexity of real-world motion, interactions, and dynamics make such implicit learning difficult Bansal et al. (2025).

In contrast, traditional graphics approaches Hu et al. (2019a; 2018; 2019b); NVIDIA (2019); Liu et al. (2024) predefine physical parameters for a scene using simulation engines, enabling them to render results that are inherently physically accurate. However, manually specifying these physical properties is often labor-intensive and becomes impractical as scene complexity increases. As a result, such methods are typically constrained to simple physical scenarios, such as perfectly elastic collisions Kang et al. (2024) and basic rigid body dynamics Liu et al. (2024). The capbility to scale to complex real-world environments, where physical effects emerge from intricate motion, interactions, forces, collisions, and environmental context, remains an open challenge.

In this work, we propose DiffPhy, a novel framework that enables video diffusion models to generate physically accurate and visually compelling videos from arbitrary user prompts. We incorporate an Multimodal Large Language Model (MLLM) Bansal et al. (2025) to verify videos against physical rules reasoned by LLM Achiam et al. (2023). The verified results, which is represented as textual tokens, is then converted into a continuous score signal, which is used to backpropagate physics-informed loss functions and resolve failed facts via attention injection. Specifically, our method begins by using an LLM Achiam et al. (2023) to infer a rich physical context from the input text, producing physical attributes, relevant physical phenomena, and an enhanced prompt. The extracted attributes and identified phenomena are then used to construct a set of physical rules for verification. We then fine-tune a pre-trained video diffusion model Wan (2025) using an enhanced prompt to make it physics-aware. Verified physical rules are incorporated to guide the gradient updates of the diffusion transformer, enforcing physical commonsense, semantic consistency, and alignment with expected physical phenomena. To enhance control over complex physical interactions, failure phenomena are encoded through an auxiliary module that injects targeted attention into challenging cases. We also introduce a new curated dataset of real videos covering diverse physical phenomena, as existing datasets are often synthetic, limited in scale, and designed for evaluation Bansal et al. (2024b; 2025); Liu et al. (2024); Motamed et al. (2025), making them unsuitable for effective fine-tuning. Our proposed dataset enables the model to be trained on complex real-world scenarios, boosting its generative capabilities.

Our contribution can be summarized as follows: **(i)** We introduce DiffPhy, a framework that enables text-to-video diffusion models to generate physically accurate and visually realistic videos from arbitrary prompts. **(ii)** We propose a novel training strategy that uses physical rules to guide gradient updates and inject attention into failure cases. This is achieved in a generalizable manner by integrating MLLM-based verification with LLM-driven physical reasoning. **(iii)** We introduce a curated dataset of real-world videos covering diverse physical scenarios to support effective training for physics-aware video generation. **(iv)** DiffPhy achieves state-of-the-art performance in generating physically plausible and semantically coherent videos across diverse scenarios.

## 2    RELATED WORK

**Text-to-Video Generation**. Diffusion-based text-to-video generation is undergoing a rapid revolution, with improvements in quality Agarwal et al. (2025), controllability Zhang et al. (2025c), efficiency Xi et al. (2025); Zhang et al. (2025b), extrapolation Zhao et al. (2025); Dalal et al. (2025), etc. Recent advancements Yang et al. (2024b); Kong et al. (2024); Wang et al. (2025) leverage the scalable Diffusion Transformer Peebles & Xie (2023) (DiT) present better synthesis quality and diversity. Closed-source models, such as SORA OpenAI (2024), Pika Pika, and Kling KlingAI

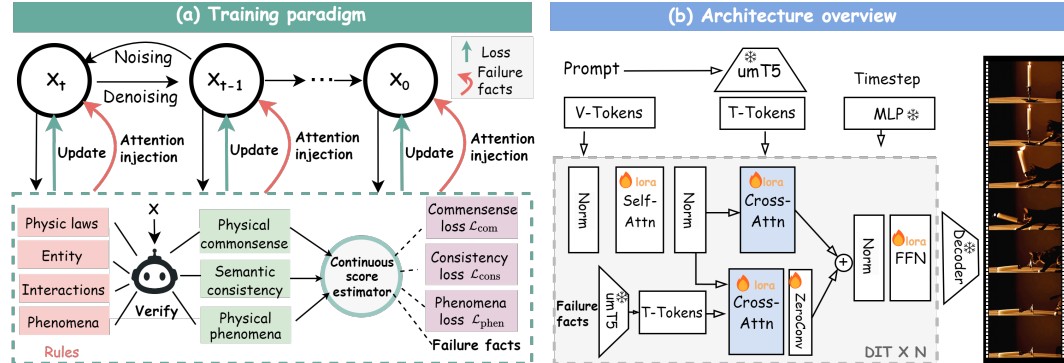

Figure 2: We present DiffPhy with (a) a training paradigm and (b) an architectural overview. Figure (a) illustrates how we incorporate verified physical rules to guide gradient updates and attention injection on the latent variables during video diffusion steps. Figure (b) illustrates the network architecture and visualizes the attention injection mechanism.

(2024), have demonstrated their impressive capabilities in video production. However, these existing methods implicitly learn the physical principles from data distribution and often neglect real-world physical laws Bansal et al. (2025), focusing more on visual aspects rather than underlying physical dynamics Kang et al. (2024).

**Video Physics Reasoning**. Existing work has used physics simulators and engines for tasks such as rigid body dynamics, fluid simulation, and deformable object modeling Hu et al. (2019a; 2018; 2019b); NVIDIA (2019); Liu et al. (2024). Recent methods Davis et al. (2015); Li et al. (2023; 2020); Wu et al. (2017; 2016); Xia et al. (2024); Xu et al. (2019) extract physical representations from visual inputs via neural networks for downstream reasoning or simulation. While effective in domains like graphics, robotics, and scientific computing, these approaches often rely on rule-based, handcrafted solvers, which limits their expressiveness, scalability, and requiring extensive manual tuning.

**LLM in T2V Generation**. LLMs have been widely used for prompt refinement in text-to-image and text-to-video generation, helping interpret prompts and guide layout planning Carreira et al. (2018); Huang et al. (2025); Lian et al. (2023); Lin et al. (2023b); Wu et al. (2024); Yang et al. (2024a); Zhu et al. (2024). Methods like PhyT2V Xue et al. (2024) refine prompts iteratively, but they operate solely in the text domain and do not improve the generation model itself. Recent work has proposed physical evaluation benchmarks that assess physical plausibility using multimodal LLMs Bansal et al. (2025); Meng et al. (2024), leveraging models such as VideoCon Bansal et al. (2024a), GPT-4o Achiam et al. (2023), LLaVA Lin et al. (2023a), and InternVideo Wang et al. (2024).

## 3 METHOD

We present DiffPhy, a generic framework that enables diffusion models to generate physically-correct and photo-realistic videos. Our core idea is to (1) Reason about the physical context and converting it into continuous signals for supervision. (2) Fine-tuning the diffusion model to become physics-aware by ensuring the generated physical phenomena align with the reasoned results. An overview of our method is shown in Figure 2. In the following, we first describe physical context reasoning (Section 3.1) and then discuss how to train the diffusion model to produce physically-correct videos (Section 3.2).

### 3.1 PHYSICAL CONTEXT REASONING

User prompts are often simple and incomplete, without detailed descriptions about the physical effects associated with the event/action. Existing methods Wan (2025); OpenAI (2024); Veo2 (2024); Yang et al. (2024b); Blattmann et al. (2023) implicitly interpret physics from such text by exhaustively training over a large data corpus, which is known to be difficult and costly Bansal et al. (2025). Instead, we leverage a pre-trained LLM to reason physical context and obtain three types of text descriptions: (1) a list of *physical attributes* that capture roles (cause/effect), interactions, and physics;(2) a list of *physical phenomena* associated with the event. (3) an *enhanced prompt* with scene and physical details.

**Physical attributes reasoning.** To extract physical attributes from text, we prompt a pretrained LLM using chain-of-thought (CoT) Wei et al. (2022). Specifically, We ask LLM to effectively identify event semantics including underlying forces (e.g., gravity, friction), kinematic relationships (e.g., constant velocity, acceleration), and interaction rules (e.g., elastic vs. inelastic collisions). We then parse the LLM output into structured machine-readable rules, creating a coherent physical description of the target scene. In practice, the user prompt might not be complete and can sometime omit important entities or interactions. To address this, we extend the CoT reasoning into a procedural process through four detailed steps: (1) identifying the key physical principles involved; (2) determining which entities initiate actions and which are affected; (3) reasoning about how these entities interact; and (4) noting any resulting physical phenomena. As a result, we obtain a list of physical attributes that are both comprehensive and valid. The detailed process is described in Appendix.

**Physical phenomena reasoning.** We also interested in obtaining a list of physical phenomena associated with the target event. These physical phenomena are facts caused by the changes of the event state, such as an action of an entity and a consequence of a physical law. In case of "a candle falling to its side", one fact might be "The candle's flame is taller when burning intensely" and "changes position from upright to lying down". The full procedure used to obtain such list is included in Appendix. These physical phenomena depict the facts to be expected in the generated videos and therefore can be used to guide the training process. For example, one can reward diffusion model when the fact is met and penalize it when the fact is missed or wrong.

Building on the extracted physical attributes and phenomena, We allow LLM to introduce new entities for logical consistency. For example, if a user prompt describes "a candle falls" in Figure 2, the LLM will also hallucinate an external force (e.g., from a cat) to make the event physically plausible. The resulting text prompt, enhanced with physical context and narrative clarity, is then used as input to guide video generation.

## 3.2 PHYSICS-AWARE MODEL TRAINING

The major challenge of physics-aware training lies in developing a diffusion model that can effectively leverage the rich physical context from text descriptions to precisely generate the desired effects. This indicates that the diffusion model needs to (1) comply with specific physical rules implied in the text prompt; (2) respect the commonsense as a natural video and (3) produce desired content adhering to the text. Ideally, this can be accomplished by optimizing the diffusion model towards some metrics that evaluate how well a generated video is aligned with specific physical effects. Yet to the best of our knowledge, there is no such measurement in the literature. To bridge this gap, we propose a three step procedure to solve this problem. (1) Physical context verification: We assess intermediate latent variables against the previously reasoned physical context across three dimensions, including physical commonsense, semantic consistency, and alignment with expected physical phenomena. (2) Continuous score estimation: The verification results are converted into continuous scores, enabling stable and differentiable supervision for training. (3)Failure-Aware Refinement: Attention is injected toward failure cases by identifying mismatches, refining the model's latent representations accordingly, and re-evaluating to ensure improved performance on challenging scenarios.

**Continuous score estimator.** Due to the inherent noise and uncertainty in LLM-generated responses, we stabilize supervision by filtering out invalid word tokens and retain only those that correspond to the valid scores. Considering each word is linked to a text token, we take the valid text token set for score $s$ as $\Omega_s$. This turns the problem into computing the expectation of these token probabilities. We first get the logits for valid tokens, apply a softmax to turn them into probabilities, and then calculate a weighted average based on the score to which each text token belongs. The final expected score is calculated as.

$$\mathbb{E}(s) = \sum_{s \in \Phi_s} s \cdot p(s), \quad \text{where } p(s) = \sum_{t \in \Omega_s} \frac{e^{z_t}}{\sum_{s \in \Phi_s} \sum_{k \in \Omega_s} e^{z_k}}, \tag{1}$$

and $z_t$ corresponds to the logit associated with the word $t$ predicted by the MLLM. For example, $\Phi_s = \{1, 2, 3, 4, 5\}$ is the set of possible scores. $\Omega_s$ is the group of tokens that represent the score $s$. For instance, the tokens "1" and "one" can both be associated with the score $s = 1$. This method gives a stable and smooth score for model training.

**Physical context verification.** We will describe about the evaluation and backpropagation details of physical phenomena alignment, physical commonsense, and semantic consistency, respectively.

*(1) Physical phenomena alignment.* It is essential to make sure that the diffusion model follows the physical principles when generating a target action/event. This is conceptually difficult to verify as the physical rules are abstract. To overcome this, we instead validate whether the generated results contain the desired facts listed in the physical phenomena, as a way to indirectly request the diffusion model to respect the underlying physical principles. Specifically, during training, at a sampled timestep $t$, we decode the predicted latent clip $x_t \in R^{m \times c \times h \times w}$ with $m$ frames into a pixel space video $v_t$. Then, we employ an MLLM Bansal et al. (2025) to evaluate the alignment between $v_t$ and each fact $f_i$ in the physical phenomena $\mathbf{F} = \{f_1, ..., f_n\}$, one by one. For each fact $f_i$, the output of MLLM can be processed as the probability vector of integer score $s_i$. Here, $s_i \in \{0, 1\}$, indicating the fact is "not matched" or "matched" in $v_t$. This allows us to define a loss function to encourage the diffusion model to produce "matched" results and penalize "unmatched" results. Specifically, for a fact $f_i$, we can compute an expected score $\mathbb{E}(s_i)$ by weighted averaging over all scores as computed by Eq.(1), where $\Phi_s = \{0, 1\}$. Given the expected score, we then define the loss function as $\mathcal{L}_{\text{phen}} = \sum_{f_i \in \mathbf{F}} ||\mathbb{E}(f_i) - 1||_2^2$. Training the model with $\mathcal{L}_{\text{phen}}$ encourages the diffusion model to generate the desired physical effects.

*(2) Physical commonsense.* We also introduce a physical commonsense loss $\mathcal{L}_{\text{com}}$ to evaluate the overall physical plausibility of the scene. We use the same multi-modal LLM again to assign an integer-valued plausibility score $s_{com}$ to $v_t$, ranging from 1 to 5, where 5 indicates the highest plausibility. Similar to $\mathcal{L}_{phen}$, we compute an expected score $s_{\text{com}}$ over all possible values (*i.e.* $\{1,2,3,4,5\}$) using Eq. (1). Then, the physical commonsense loss can be defined as a mean squared error (MSE): $\mathcal{L}_{com} = ||\mathbb{E}(s_{\text{com}})/\tau - 1||_2^2$, where $\tau = 5$ is a normalization constant. This loss encourages the diffusion model to generate videos that meet physical commonsense.

*(3) Semantic consistency.* To ensure the generated content still follows the input prompt, we introduce a semantic consistency loss. Again, we prompt the pretrained multi-modal LLM to give scores from 1 to 5 for evaluating the semantic consistency between $v_t$ and the input text prompt. The semantic consistency loss has the same format as physical commonsense loss, *i.e.* $\mathcal{L}_{sem} = ||\mathbb{E}(s_{\text{sem}})/\tau - 1||_2^2$, where $\tau = 5$ is a normalization constant, $s_{\text{sem}}$ is the expected score over all possible values.

**Failure-Aware Refinement.** It is equally important for the diffusion model to resolve any failed or incorrect facts. For a "not matched" fact $f_i$, we introduce an additional module to inject attention and guide the diffusion model to better understand the failure or missed cases during generation. This module encodes $f_i$ into text embeddings and conditions the diffusion model via a separate cross-attention layer in each DIT block, as shown in Figure 2-(b). To stabilize the training, we copy the original cross-attention module and connect it with the original model via a zero-initialized convolution layer (ZeroConv) in a way similar to Zhang et al. (2023). To reduce training cost, we apply a LoRA adapter Hu et al. (2022) to the cloned attention module and fine-tune only the adapter. More implementation details can be found in the Appendix.

**Training details.** The final loss is computed by combining the original denoising score matching $\mathcal{L}_\epsilon$ objective Ho et al. (2020) with the three newly proposed MLLM-based loss functions, *i.e.*, phenomena loss $\mathcal{L}_{\text{phen}}$, physical commensense loss $\mathcal{L}_{\text{com}}$ and semantic consistency loss $\mathcal{L}_{\text{sem}}$. Formally,

$$\mathcal{L} = \mathcal{L}_\epsilon + \beta(\mathcal{L}_{\text{phen}} + \mathcal{L}_{\text{com}} + \mathcal{L}_{\text{sem}}), \tag{2}$$

where $\beta$ is a balancing hyperparameter. As shown in Figure 2, we freeze the original DIT blocks and train only LoRA layers to reduce training cost. When a failure fact occurs, we sample the same diffusion step again with the failure fact-based guidance injected. We also randomly perform a training step without activating the injection branch. This allows diffusion model to still generate high-quality videos from solely text input. More implementation details are provided in the Appendix.

## 4 EXPERIMENTS

We encourage readers to watch the supplemental video, which provides a comprehensive comparison. **HQ-Phy Dataset.** We curate a dataset of approximately 8,000 videos selected from the VIDGEN-1M dataset Tan et al. (2024), which includes the video, caption, enhanced prompts, and extracted physical phenomena labels. We filter the dataset to exclude cartoon and other unrealistic scenes. To extract the enhanced prompts and detailed physical phenomena from this training dataset, we use DeepSeek-R1-32B Guo et al. (2025) as a cost-effective alternative to GPT.

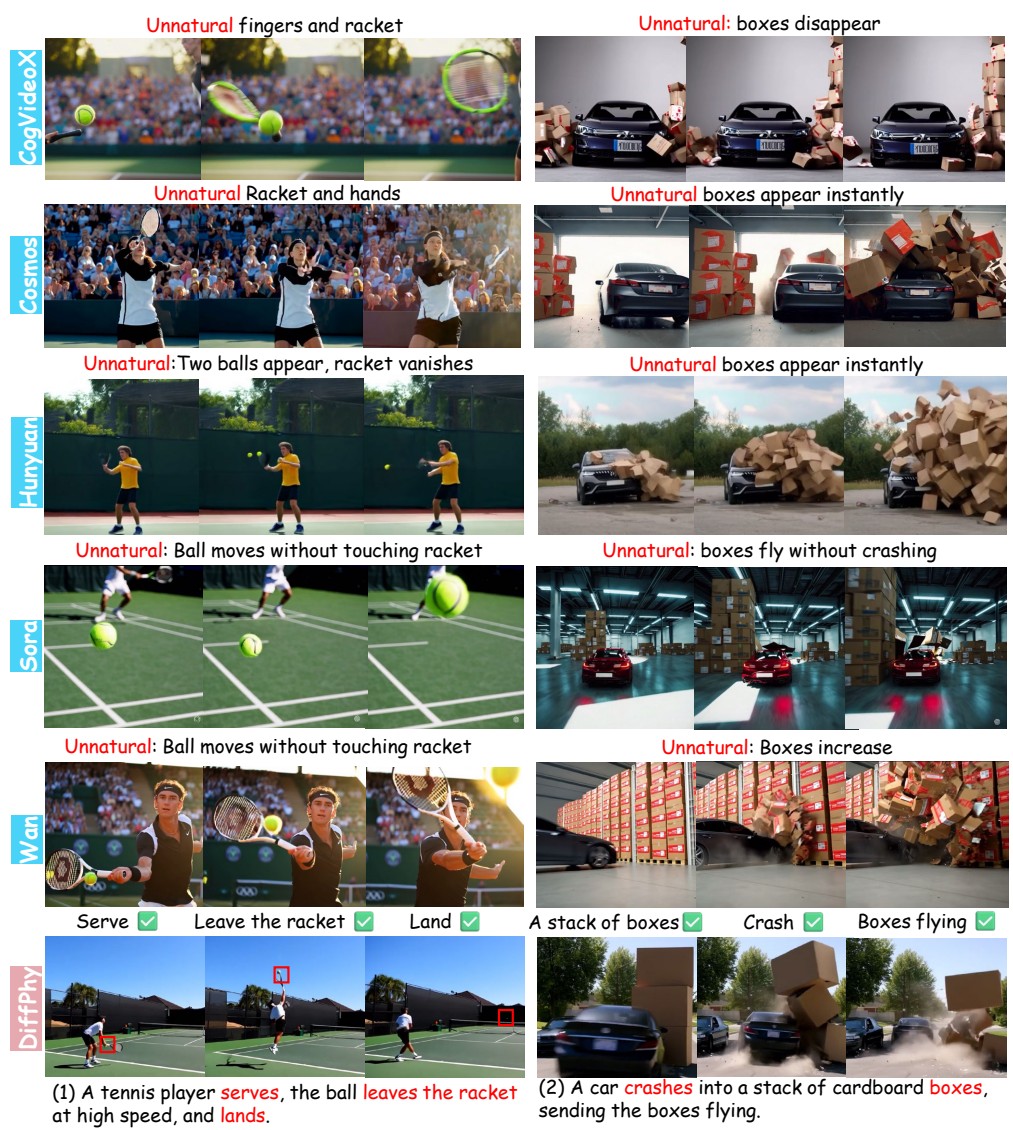

Figure 3: Qualitative comparison with T2V models on the VideoPhy2. We show two challenging cases, *i.e.*, sports and box collapse, where our results are more natural and description-consistent.

**Evaluation benchmarks.** In our experiments, we utilize two datasets specifically designed to evaluate physical commonsense reasoning in text-to-video generation. *VideoPhy2* Bansal et al. (2025) provides a collection of 590 human-verified captions that describe realistic interactions between various physical entities, serving as a testbed for assessing whether generated videos reflect plausible physics. Additionally, we incorporate *PhyGenBench* Meng et al. (2024), a benchmark constructed to probe the physical awareness of T2V models. It comprises 160 designed prompts spanning four fundamental domains of physics: mechanics, optics, thermal processes, and material properties.

**Metrics.** To ensure a comprehensive evaluation of our generated videos, we adopt a multi-faceted assessment strategy combining model-based metrics and human judgment. Firstly, we adopt the evaluation metrics from the PhyGenBench benchmark Meng et al. (2024). Following PhyGenBench Meng et al. (2024), we report key phenomena detection, physical order verification, and average scores to indicate overall quality using GPT-4o Achiam et al. (2023) and open-source models, *i.e.*, CLIP Radford et al. (2021), InternVideo2 Wang et al. (2024), and LLaVA Lin et al. (2023a). Second, we use the VideoCon-Physics evaluator from VideoPhy2Bansal et al. (2025), which reports two metrics: physical commonsense (PC) and semantic adherence (SA), each scored from 1 to 5. PC evaluates physical plausibility, while SA measures alignment with the input prompt. Following VideoPhy2Bansal et al.

| Model | Physical domains(↑) | | | | Average |
|-------|-----------|--------|---------|----------|---------|
| | Mechanics | Optics | Thermal | Material | |
| CogVideoX-2B | 0.38 | 0.43 | 0.34 | 0.39 | 0.37 |
| CogVideoX-5B | 0.43 | 0.55 | 0.40 | 0.42 | 0.45 |
| Open-Sora V1.2 | 0.43 | 0.50 | 0.44 | 0.37 | 0.44 |
| Lavie | 0.40 | 0.44 | 0.38 | 0.32 | 0.36 |
| Vchitect 2.0 | 0.41 | 0.56 | 0.44 | 0.37 | 0.45 |
| Pika | 0.35 | 0.56 | 0.43 | 0.39 | 0.44 |
| Kling | 0.45 | 0.58 | 0.50 | 0.40 | 0.49 |
| Wan | 0.36 | 0.53 | 0.36 | 0.33 | 0.40 |
| DiffPhy (ours) | **0.53** | **0.59** | **0.58** | **0.46** | **0.54** |

Table 1: T2V comparisons on PhyGenBench.

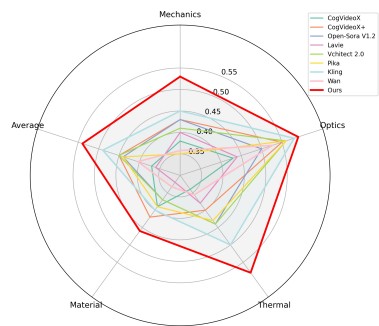

Figure 4: Visualization of Table 1.

| Dimension | Methods | Physical metrics (↑) | | Overall Quality | | Average |
|-----------|---------|-----------|-------|--------|------|---------|
| | | Phenomena | Order | GPT-4o | Open | |
| Mechanics | Kling | 0.608 | 0.454 | 0.533 | 0.508 | 0.450 |
| | Wan | 0.608 | 0.400 | 0.442 | 0.483 | 0.358 |
| | DiffPhy | **0.733** | **0.525** | **0.617** | **0.567** | **0.533** |
| Optics | Kling | 0.753 | 0.550 | 0.660 | **0.660** | 0.580 |
| | Wan | 0.760 | 0.563 | 0.587 | 0.573 | 0.527 |
| | DiffPhy | **0.833** | **0.660** | **0.667** | 0.647 | **0.587** |
| Thermal | Kling | 0.611 | 0.372 | 0.544 | 0.567 | 0.500 |
| | Wan | 0.500 | 0.272 | 0.378 | 0.500 | 0.356 |
| | DiffPhy | **0.700** | **0.578** | **0.622** | **0.656** | **0.578** |
| Material | Kling | 0.617 | 0.342 | 0.492 | 0.483 | 0.400 |
| | Wan | 0.575 | 0.254 | 0.392 | 0.417 | 0.333 |
| | DiffPhy | **0.725** | **0.433** | **0.575** | **0.500** | **0.458** |

Table 2: PhyGenBench evaluation of phenomena detection, physical order, GPT-4o and open-source models.

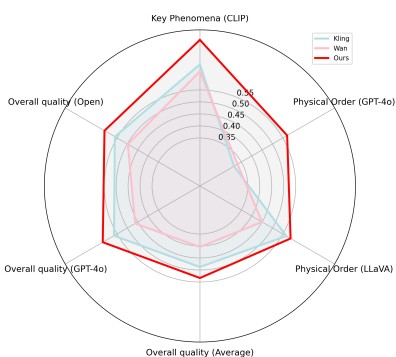

Figure 5: Average metrics of Table 2.

(2025), we set the overall score to 1 if both PC and SA are $\geq 4$, and 0 otherwise. Finally, we conduct a gold-standard human evaluation via blind review, where three annotators independently score each video on semantic adherence and physical commonsense.

**Baselines.** We apply DiffPhy on Wan2.1-14B Wan (2025), a DiT-based open-source T2V models, and compare it to eleven open or close-source T2V models, including Wan2.1-14B Wan (2025), CogVideoX-5B Yang et al. (2024b), VideoCrafter2 Chen et al. (2024), HunyuanVideo-13B Kong et al. (2024), Cosmos-Diffusion-7B Agarwal et al. (2025), OpenAI Sora OpenAI (2024), Luma Ray2 LumaAI (2025), LaVie Wang et al. (2023), Vchitect Fan et al. (2025), Pika Pika, and Kling KlingAI (2024). Although DiffPhy supports various backbones, we use Wan2.1-14B for illustration.

### 4.1 PHYGENBENCH EVALUATION

Table 1 and Figure 4 present the performance of our model compared with a range of existing methods, including CogVideoX (2B/5B) Yang et al. (2024b), OpenSora OpenAI (2024), LaVie Wang et al. (2023), Vichitect Fan et al. (2025), Pika Pika, Kling KlingAI (2024), and Wan 2.1-14B Wan (2025). The results reported in Table 1 are sourced from the official PhyGenBench leaderboard Meng et al. (2024), with the results for Wan 2.1-14B reproduced using its publicly released code and model Wan (2025). Our DiffPhy not only outperforms Wan by a substantial margin across all metrics but also achieves the highest average score and the best performance in all of the dimensions, demonstrating the effectiveness of our proposed strategies.

**Physical phenomena:** We further validate the ability of our method to capture detailed physical phenomena. Table 2 and Figure 5 compare the performance of our model, the baseline Wan, and the second-ranked closed-source model Kling KlingAI (2024) on aspects including key phenomena (phenomena), physical order (order), and overall quality. The overall quality is indicated by the average results from GPT-4o Achiam et al. (2023) and open-sourced models (Open), *i.e.*, CLIP Radford et al. (2021), LLaVA Lin et al. (2023a), and InternVideo2 Wang et al. (2024). Our model achieves the highest scores in both key phenomena and physical order, with strong average performance across four physics-related categories: Mechanics, optics, thermal, and material properties. These results

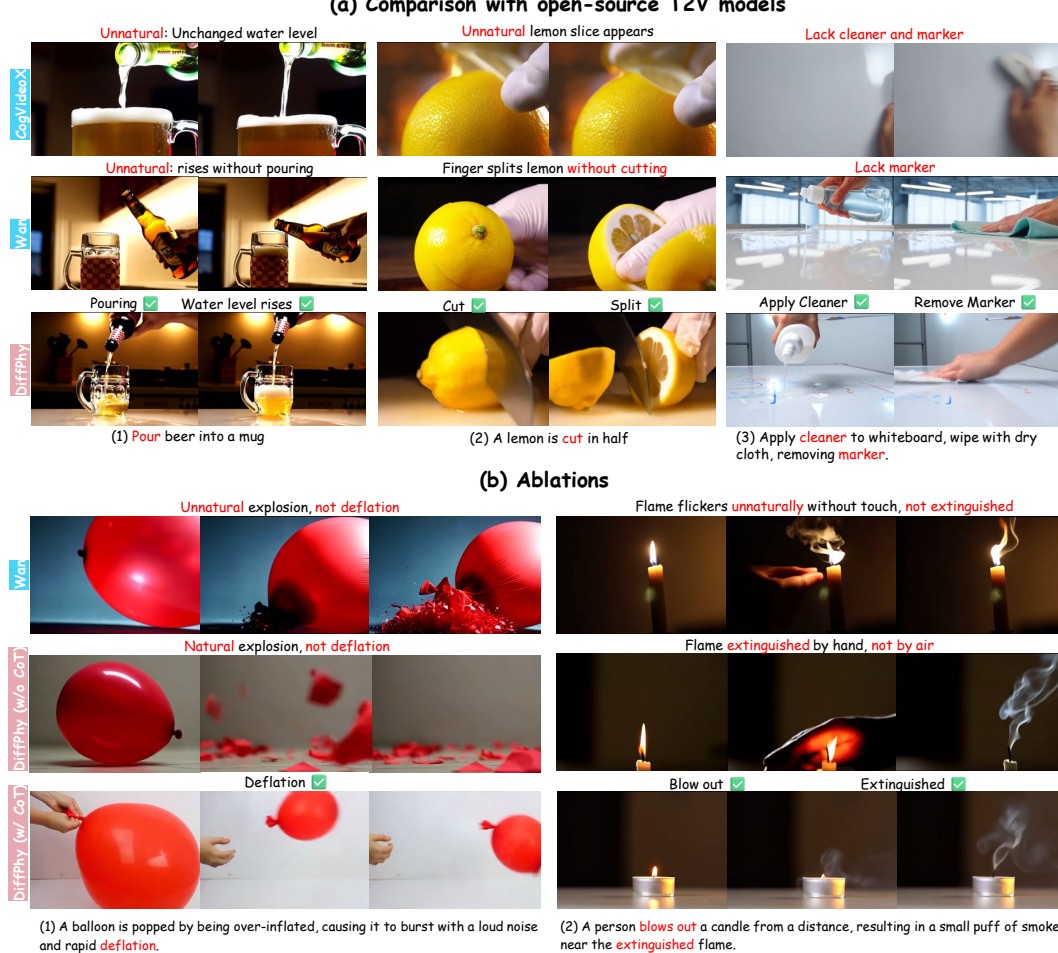

Figure 6: Visualization of (a) Open-source T2V model comparison on physical activities, and (b) ablation study on physical interactions. All prompts and compared videos are from VideoPhy2.

| Methods | Model-based | | | | Human-based | | | |
|---|---|---|---|---|---|---|---|---|
| | Hard | Activity | Interaction | Overall | Hard | Activity | Interaction | Overall |
| Videocrafter | 0.054 | 0.201 | 0.229 | 0.205 | 0.131 | 0.101 | 0.131 | 0.105 |
| Sora | 0.053 | 0.244 | 0.200 | 0.217 | 0.267 | 0.222 | 0.267 | 0.233 |
| Ray2 | 0.067 | 0.259 | 0.204 | 0.234 | 0.185 | 0.210 | 0.185 | 0.203 |
| Hunyuan | **0.096** | 0.255 | 0.312 | 0.267 | 0.159 | 0.176 | 0.159 | 0.172 |
| Cosmos | 0.034 | 0.219 | 0.223 | 0.229 | 0.274 | 0.226 | 0.274 | 0.241 |
| cogvideo | 0.051 | 0.258 | 0.236 | 0.261 | 0.000 | 0.246 | 0.261 | 0.250 |
| Wan | 0.034 | 0.210 | 0.255 | 0.225 | 0.219 | 0.315 | 0.362 | 0.326 |
| DiffPhy (w/o CoT) | 0.045 | 0.245 | 0.255 | 0.247 | 0.386* | 0.519* | 0.400* | 0.489* |
| DiffPhy (w/ CoT) | 0.084 | **0.266** | **0.317** | **0.281** | **0.421*** | **0.526*** | **0.556*** | **0.533*** |

Table 3: Comparison of model-based and human-based scores on the VideoPhy2 dataset. ∗ indicates our human-based scores; others are from the Videophy2 benchmark.

indicate that our model not only identifies key physical phenomena accurately but also generates videos that follow coherent physical progression.

## 4.2 VIDEOPHY BECHMARK

Table 3 and Figure 3 demonstrate that our method DiffPhy, significantly outperforms the existing T2V approaches on the VideoPhy2 dataset. Compared to Wan, our model shows notable improvements in average performance across all subcategories, including sports and physical activities (Activity), object interactions (Interaction), and challenging physical scenarios (Hard). We evaluate our method on the VideoPhy2 benchmark, following the protocol outlined in Bansal et al. (2025). To ensure a

| Methods | Hard | | | Activity | | | Interaction | | | Average | | |
|---|---|---|---|---|---|---|---|---|---|---|---|---|
| | SA | PC | Overall | SA | PC | Overall | SA | PC | Overall | SA | PC | Overall |
| CogVideoX | 2.018 | 2.193 | 0.018 | 2.452 | 2.415 | 0.141 | 2.311 | 2.467 | 0.022 | 2.417 | 2.428 | 0.111 |
| Wan | 3.439 | 3.421 | 0.298 | 3.489 | 3.504 | 0.378 | 3.578 | 3.378 | 0.311 | 3.511 | 3.472 | 0.361 |
| DiffPhy (w/o CoT) | 3.456 | 3.439 | 0.386 | 3.793 | 3.704 | 0.519 | 3.556 | 3.489 | 0.400 | 3.733 | 3.650 | 0.489 |
| DiffPhy (w/ CoT) | **3.702** | **3.491** | **0.421** | **3.800** | **3.770** | **0.526** | **3.867** | **3.822** | **0.556** | **3.817** | **3.783** | **0.533** |

Table 4: Our human-based scores on VideoPhy2 dataset. The scores of compared methods are re-evaluated by us.

| Method | Training | | Inference | | Model-based | | | Human-based | | |
|---|---|---|---|---|---|---|---|---|---|---|
| | $\mathcal{L}_{com}$&$\mathcal{L}_{sem}$ | $\mathcal{L}_{phen}$ | Enhance Prompt | Iterative | Activity | Interaction | Overall | Activity | Interaction | Overall |
| Wan | × | × | × | × | 0.210 | 0.255 | 0.225 | 0.361 | 0.378 | 0.311 |
| #1 | ✓ | × | × | × | 0.238 | 0.267 | 0.245 | 0.382 | 0.441 | 0.386 |
| #2 | ✓ | ✓ | × | × | 0.245 | 0.255 | 0.247 | 0.400 | 0.519 | 0.489 |
| #3 | ✓ | × | ✓ | × | 0.259 | 0.276 | 0.270 | 0.478 | 0.523 | 0.492 |
| #4 | ✓ | ✓ | ✓ | × | **0.266** | **0.317** | **0.281** | **0.526** | **0.556** | **0.533** |
| #5 | ✓ | ✓ | ✓ | ✓ | 0.273 | 0.323 | 0.289 | 0.535 | 0.560 | 0.542 |

Table 5: Ablation results on VideoPhy2. $\mathcal{L}_{com}$, $\mathcal{L}_{sem}$, $\mathcal{L}_{phen}$ denote physical commonsense loss, semantic consistency loss, and physical phenomena loss, respectively; *Iterative* refers to an optional second-pass refinement by injecting the failure phenomena during inference.

fair comparison, we report results both with and without CoT enhanced prompt. While our baseline already surpasses Wan, incorporating CoT further boosts the performance, achieving the highest average scores across all video categories. Figure 6 visualizes typical videos generated by our model with and without CoT enhanced prompt, compared to the baseline Wan model. This demonstrates DiffPhy 's ability to generate coherent and plausible videos. We also observe discrepancies between model-based and human-based ratings, which is also mentioned in Bansal et al. (2025). The root of this discrepancy is that human evaluation serves as the definitive standard for physical plausibility, a role that model-based metrics only partially fulfill. Results for the compared methods, including both model and human evaluations, are taken from the original VideoPhy2 leaderboard Bansal et al. (2025). Our own human evaluation results are provided in Table 4.

**Human evaluation.** We conduct a human evaluation on two dimensions consistent with Video-Phy2 Bansal et al. (2025), including Semantic Alignment (SA) and Physical Commonsense (PC). Table 4 compares our model with the top two models on the Videophy2 human-annotated leaderboard, CogVideoX Hong et al. (2022) and Wan 2.1-14B Wang et al. (2025). Our model outperforms Wan across all three evaluation dimensions, even without CoT. CoT further boosts the performance across all dimensions.

### 4.3 ABLATIONS

We conduct ablation studies to evaluate the effectiveness of each design choices. Results are reported in Table 5. We have the following findings. First, incorporating MLLM-based loss functions yields substantial gains over the Wan Wan (2025) baseline on both automatic and human evaluation. Second, adding the attention injection to resolve failure facts with phenomena loss ($\mathcal{L}_{phen}$) further improves performance. Third, incorporating CoT prompting enhances prompt clarity and reasoning, resulting in additional improvement. Visual results on the effects of CoT are reported in Figure 6. During inference, our DiffPhy framework supports an optional two-pass inference by correcting failure cases from previous pass via attention injection, which yields additional performance improvements. Because it introduces runtime overhead, for fair comparison, we only report results using one-pass inference for the main experiments.

## 5 CONCLUSION

We introduce DiffPhy, a novel framework that enables diffusion models to generate physically-correct and realistic videos from arbitrary user prompts. This is achieved by leveraging Large Language Models (LLMs) to reason physical context from the input prompt and use it to guide the diffusion process. We then finetune the diffusion model to make use of this context using a Multimodal LLMs (MLLMs) as supervisory signals. Additionally, we provide a curated dataset of real-world videos covering a wide range of physical phenomena. Experiments on public benchmarks demonstrate that DiffPhy achieves state-of-the-art performance for physics-aware video generation.

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

## A  OVERVIEW OF APPENDIX

In this appendix, we first present our demo video in Section B. Next, we describe the implementation details of our experiments in Section C. Section D provides a detailed overview of the prompts and generated prompts used in our Chain-of-Thought (CoT) process. Additional qualitative comparisons are visualized in Section E. In Section F, we introduce the PhyHQ dataset, including its distribution and representative video examples. Finally, we discuss the limitations of our approach and outline directions for future work in Section G.

## B  VIDEO DEMONSTRATION

We encourage readers to watch the supplemental video, which showcases visual results and comparisons, providing a more comprehensive illustration of both the physical realism and generation quality of DiffPhy.

## C    IMPLEMENTATION DETAILS

**Experiment setup:**    All experiments are conducted using four NVIDIA H100 GPUs with a batch size of 4. We fine-tune all models for 10 epochs, using Wan 2.1-14B Wan (2025) as the base model, following its training paradigm and data preprocessing setup. Specifically, we fine-tune the LoRA adapter on top of the Wan 2.1-14B backbone, using a LoRA alpha of 32 and a rank of 16. The hyperparameter $\beta$ of the training objective is empirically set to 0.1. To maintain training flexibility, we enable the diffusion model to occasionally skip applying the physical phenomena violation guidance, with a probability of 0.1. Thus, the model can generate high-quality videos even without the guidance of physical phenomena.

**LLM:**    We use LLM to perform Chain-of-Thought (CoT) reasoning, to generate enhanced prompt and physical phenomena. Specifically, we employ GPT O4-mini Inc. (2024) for CoT during inference. For the large-scale training dataset, we use DeepSeek-R1-32B Guo et al. (2025) as a cost-effective open-source option. The detailed process are described in  D.

**MLLM:**    The multimodal LLM is used to guide the model in generating physically plausible videos during training. We adopt the multimodal LLM evaluator VideoCon-Physics Bansal et al. (2025) for its lightweight, open-source design. It is built on VideoCon Bansal et al. (2024a), with CLIP Radford et al. (2021) as the visual encoder and LLaMA-7B Touvron et al. (2023) as the language backbone. For MLLM Output Processing, due to the inherent noise and uncertainty in LLM-generated responses, we apply a post-processing strategy to stabilize supervision by filtering out invalid word tokens. Specifically, we use the pretrained multimodal LLM evaluator to predict discrete score distributions: integers from 1 to 5 for semantic consistency and physical commonsense, and labels $0$ : unfollow", $1$ : follow", $2$ : "undetermined" for each physical phenomenon. Since the multimodal LLM outputs textual tokens rather than strict integers, some predictions may be noisy or invalid. We therefore filter out invalid tokens and retain only those that correspond to the valid integer labels.

**Evaluation prompts:**    When generating videos for benchmark evaluation, *i.e.*, PhyGenBench and VideoPhy, we use the same input prompts across all compared methods and DiffPhy (w/o CoT). These prompts are taken directly from the benchmarks provided by VideoPhy2 Bansal et al. (2025) and PhyGenBench Meng et al. (2024), using their augmented prompts of the original user inputs. For DiffPhy (w/ CoT), we use our own enhanced prompts generated through Chain-of-thought physical reasoning.

**Human evaluation:**    We use human evaluation results as the ground-truth metric to complement model-based evaluations. We design a blind evaluation platform and conduct a blind review process for human annotation, where each annotator is asked to assess the physical commonsense and semantic alignment of text-to-video generation results. To ensure consistency, we follow the same video sampling strategy used by Bansal et al. (2025) for evaluating Sora.

**Curation of PhyHQ dataset:**    We curate the PhyHQ dataset to enable effective finetuning of physics-aware video diffusion models. Firstly, we collect the videos selected from the VIDGEN-1M dataset Tan et al. (2024), which includes the videos and corresponding captions. Then, we further extract physical phenomena labels using the open-source DeepSeek-R1-32B Guo et al. (2025), guided by the prompt described in Section D. This serves as a cost-effective alternative to using GPT-based models. We filter out the unrealistic videos with zero-shot-classification of BART-large-mnli Lewis et al. (2020) based on the video descriptions. For dataset preprocessing, we follow the sampling and process pipeline of Wan 2.1  Wan (2025)

## D    CHAIN-OF-THOUGHT (COT)

As visualized in Figure 7, we provide illustration for CoT. Given a user prompt, we leverage a pretrained LLM to reason physical properties from the text input. We then (a) enhance the user prompt with physical context and (b) produce a list of relevant physical phenomena associated with the described event. We use the enhanced prompt to guide video generation. The phenomena list is

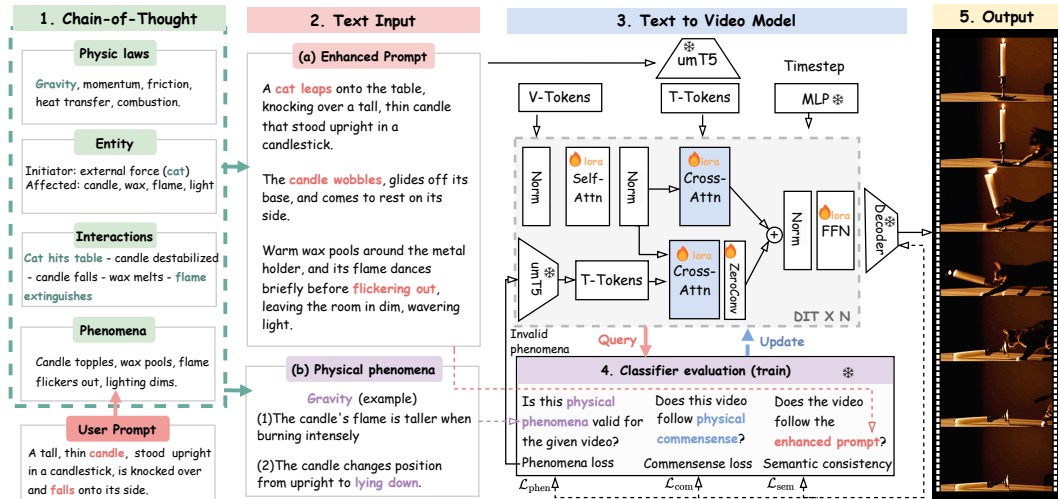

Figure 7: An overview of DiffPhy with CoT illustration

used to penalize outputs with implausible physics. A set of training objectives are proposed to jointly enforce physical correctness and semantic consistency.

## D.1 CoT Generation

During the Chain-of-Thought (CoT) reasoning process, the LLM is instructed to identify relevant physical laws, the resulting phenomena, the affected entities, and how their behavior changes as a consequence.

---

**Instruction for CoT Generation**

Analyze the scene by:
1. Identifying the key physical principles at work.
2. Determining which entities initiate actions and which are affected.
3. Examining how these entities interact.
4. Noting any observable physical phenomena that result.
5. Describe the scene vividly and clearly, capturing the key actions, objects, and atmosphere in natural language.

Ensure that all objects mentioned in the original caption are included in the output, using the same wording for each entity. Also, include any entities or individuals involved in the interaction to ensure the scenario remains logical and coherent. The output should be a natural and simple description of the observed phenomena, without the detailed physical analysis, written in plain language and limited to 70 words.

*Input scene:* A delivery drone descends into a residential courtyard, avoiding tree branches and landing precisely on a designated pad.

*Output enhanced prompt:* A compact delivery drone hovers quietly above a residential courtyard, its rotors emitting a soft hum. It skillfully avoids low-hanging branches and lands precisely on a marked pad set into the stone tiles. The matte-gray frame stands out against the backdrop of greenery and pastel buildings, completing its task in one smooth motion.

---

To clarify the expected output structure, we leverage *in-context learning*—a technique where models are provided with annotated examples within the prompt to illustrate the desired reasoning and response format—ensuring the LLM aligns its output with task-specific requirements. To maintain coherence and minimize noise in the generated responses, we instruct the LLM to consistently reuse terminology from the original prompt when referring to the same entities. We also instruct the LLM to focus more on describing observable phenomena rather than abstract physical laws in the enhanced

810
811
812
813
prompt, as phenomena are generally easier for the diffusion model to interpret and learn from than explicit rules. Additionally, to avoid overwhelming the model with excessive context, which can obscure key interactions, we explicitly limit the length of the enhanced prompt in the instructions. The prompt used for CoT generation is shown below.

814
815
816
D.2 EXTRACTING PHYSICAL PHENOMENA FROM TRAINING DATASET

817
818
819
Given the cost associated with using the GPT API, we utilize DeepSeek-R1-32B Guo et al. (2025) to extract physical phenomena from large-scale training data. We provide illustrative examples and instruct DeepSeek to generate physical rules in JSON format. The prompt is shown above.

820
821
822
823
824
Although we also prompt DeepSeek to produce various versions of enhanced prompts, we find that the original video captions generally offer higher quality. This is because DeepSeek often generates imagined scenarios that do not accurately reflect the video content. As a result, we retain the original captions for training and use only the physical rules and phenomena extracted by DeepSeek.

825
826
**Instruction for Extracting Physical Phenomena**

827
828
829
830
Task: Enhance video captions by
   (1) identifying physical rules, then
   (2) rewriting the text explicitly using those rules.
Goal: Generate 5 diverse, faithful rewrites of the original description, grounded in physics.

831
832
833
834
835
836
837
838
839
840
841
842
*Instructions*:
   1. Physical Rules Extraction: Analyze the input description for observable physical phenomena; Output JSON format: "Observation": "Physics Principle", and Wrap rules with <physics_rules></physics_rules>.
   2. Rewriting Requirements: Expand descriptions using identified physics principles; Create 5 variants with different styles that may include (but not limited to) the following features:
      (a) Naturally integrate physics terms (friction, momentum, etc.) into narrative
      (b) Vary sentence structure complexity (concise vs elaborate)
      (c) Use hybrid technical/natural language (e.g. "friction (surface adhesion)" parentheses)
      (d) Maintain original semantic core
   3. Keep faithful to original content
   4. Wrap each rewrite with <rewrite1></rewrite1> through <rewrite5></rewrite5>

843
844
845
846
847
848
849
850
851
852
853
*Example Input*: "A Segway bumps over small speed bumps, rider maintaining balance."
*Example Output*:
<physics_rules> {{ "The Segway wheels rotate.": ["Friction"], "The rider and Segway move forward.": ["Conservation of Momentum"], "The rider's shadow length changes as the sun's angle changes.": ["Reflection"], "The Segway should experience a change in speed when going over speed bumps": ["Friction"] }} </physics_rules>
<rewrite1>A Segway navigates speed bumps, its rubber tires generating friction as they rotate against the pavement. The rider leans forward, conserving momentum through the obstacle course while their elongated shadow reflects the low sun angle. Each bump momentarily reduces speed as friction increases between wheels and concrete.</rewrite1>

854
855
...

856
857
858
859
860
861
862
863
Now process: "{original_caption}"
Required Output Format:
<physics_rules>
{{JSON}}
</physics_rules>
<rewrite1>...</rewrite1>
...
<rewrite5>...</rewrite5>

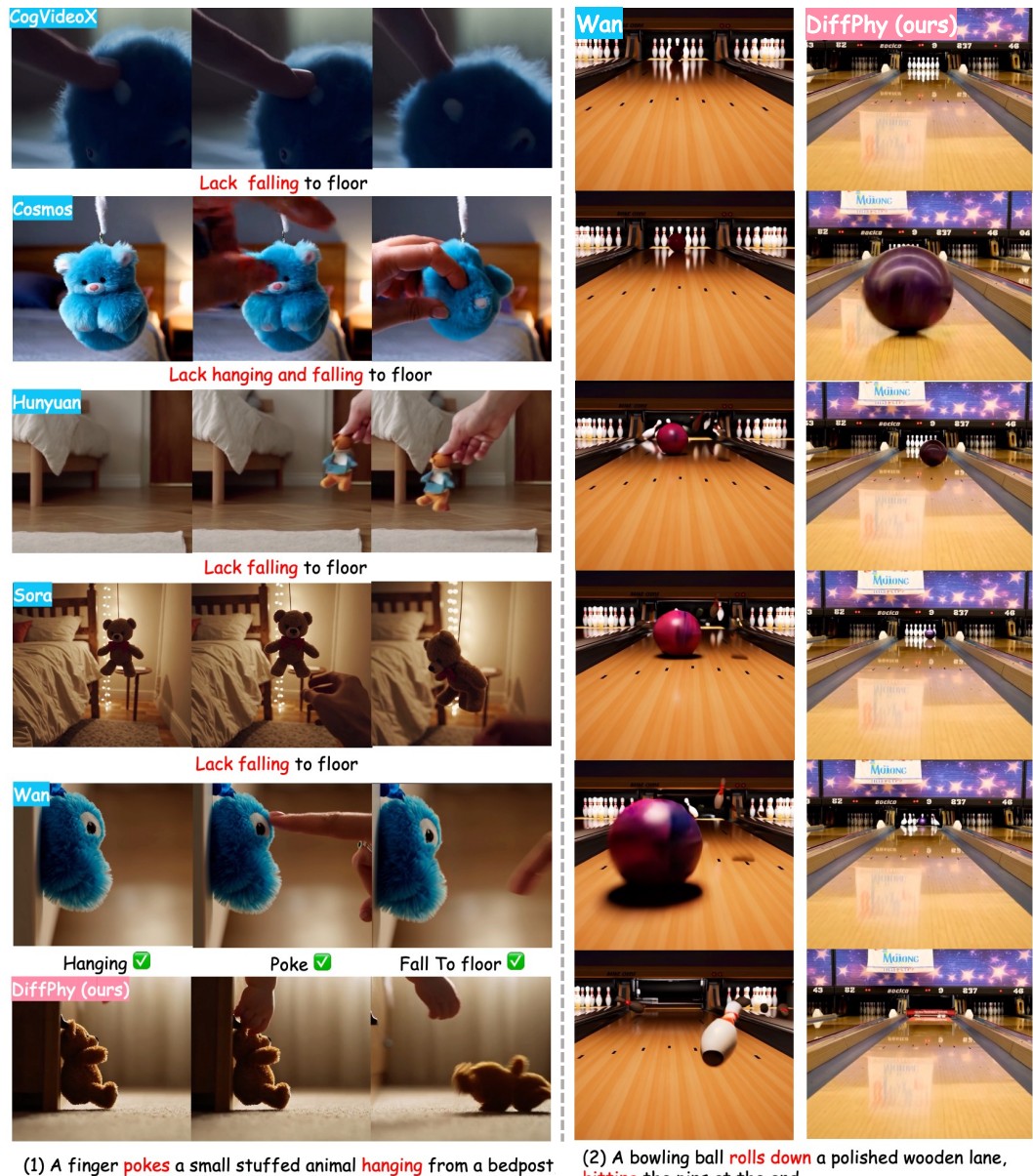

(1) A finger **pokes** a small stuffed animal **hanging** from a bedpost causing it to **fall to the floor**.

(2) A bowling ball **rolls down** a polished wooden lane, **hitting** the pins at the end.

Figure 8: DiffPhy enables physically reasonable and semantically coherent video generation in challenging cases. (1) shows the comparison to T2V models and (2) provides multi-frame comparison to Wan 2.1-14B. DiffPhy generates more semantically coherent and physically reasonable videos compared to other methods.

### D.3 COMPARISON OF ENHANCED PROMPTS

To demonstrate the effectiveness of our proposed CoT reasoning, we present a prompt comparison using the videos shown in Figure 8-(1) as an example. First, we show the original user input, which is a short descriptive sentence. Next, we include the enhanced prompt generated by VideoPhy, followed by the prompt produced by our Chain-of-Thought (CoT) method. For reference, we also provide the phenomena checklist used by VideoPhy. Finally, we provide discussion about the advantage of our enhanced prompts.

**Input:** A finger pokes a small stuffed animal hanging from a bedpost, causing it to fall to the floor.

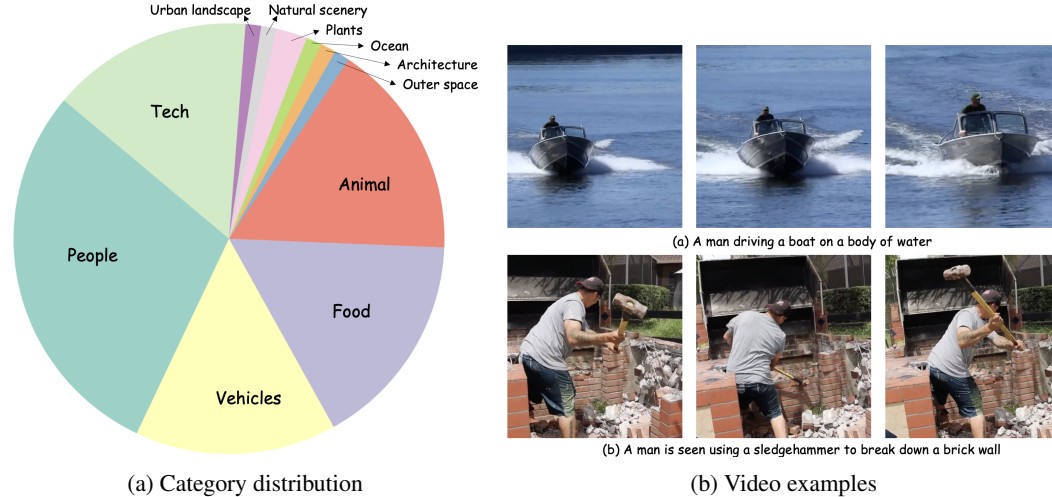

(a) Category distribution  (b) Video examples

Figure 9: Illustration of PhyHQ dataset. (a) shows the distribution of video categories and (b) shows examples of video from PhyHQ dataset.

**Enhanced prompt by VideoPhy:**  In a cozy, dimly lit bedroom, a small, plush stuffed animal dangles from a bedpost, its soft, fluffy body swaying gently in the still air. The camera captures a close-up of the toy, its vibrant blue fur and large, round eyes inviting warmth and comfort. As the scene unfolds, a human finger, adorned with a delicate ring, enters the frame, its tip gently pressing against the toy's head. The soft pressure causes the toy to sway, its head tilting slightly, evoking a playful, interactive moment. The background remains blurred, enhancing the focus on the toy and the finger, while the lighting casts a gentle glow, creating a serene atmosphere. The finger's movement is deliberate yet tender, as it continues to poke and nudge the toy, ultimately causing it to tumble down, landing softly on the floor. The camera remains static, capturing the subtle interplay of touch and the toy's descent, inviting viewers into a world of playful exploration.

**Enhanced prompt by DiffPhy:**  A slender finger reaches out and pokes a small stuffed animal hanging from a bedpost. The plush toy wobbles and tilts, its stitches rustling softly before it loses its grip and falls to the floor. The gentle thud echoes in the quiet room, where soft light filters through curtains, casting warm patterns across the carpet.

**Physical phenomena:**  *Elasticity*: The finger makes contact with the toy, causing a slight deformation of the toy's fur. *Gravity*: The toy remains attached to the shelf after being poked. *Friction*: The toy does not move significantly from its original position on the shelf after being poked.

**Discussion:**  Although VideoPhy generates rich contextual descriptions, it tends to focus more on atmospheric details rather than the physical interactions themselves. As a result, while the scenes may appear vivid, the generated videos often fail to capture key interactions—such as poking or objects falling to the floor. In contrast, the enhanced prompts produced by our CoT reasoning approach explicitly describe both the critical physical phenomena and the surrounding atmosphere. The descriptions of physical interactions ensure physical commonsense and semantic coherence, while the atmospheric details contribute to overall naturalness. We also include the physical phenomena identified by VideoPhy for reference. As shown in Figure 8, our generated video aligns well with these phenomena, demonstrating the effectiveness of our approach.

# E ADDITIONAL QUALITATIVE COMPARISONS

For qualitative comparison, we included results against existing T2V methods in the main manuscript. Here, Figure 8 provides additional comparisons between our approach and existing methods. Specifically, Figure 8(a) highlights differences in semantic coherence when evaluated against several state-of-the-art text-to-video models, including CogVideoX, Cosmos, Hunyuan, Sora, and Wan. Figure 8(b) offers a detailed frame-by-frame comparison between Wan and our DiffPhy model across six

frames. For a more comprehensive understanding and additional qualitative insights, we encourage readers to view our supplemental demo video.

## F    DETAILS OF PHYHQ DATASET

The PhyHQ dataset consists of 8,000 videos, each paired with a caption and corresponding physical phenomena, curated according to the procedure described in Section C. Figure 8 illustrates the category distribution and provides example videos from PhyHQ. The dataset spans a wide range of categories, including people, vehicles, food, animals, technology, urban landscapes, natural scenery, plants, oceans, architecture, and outer space. As illustration, we provide the detailed text prompts used in cases (a) and (b) below. (a) In the video, a man is seen using a sledgehammer to break down a brick wall. He is wearing a gray shirt, black shorts, and a black cap. The man is standing on a pile of rubble and debris, indicating that he has been working on this task for some time. The wall appears to be part of a larger structure, possibly a building or a house. The man is focused on his task, and his movements are deliberate and forceful as he swings the sledgehammer to break the bricks. The sound of the hammer hitting the bricks is loud and echoes in the surrounding area. The man's body language suggests that he is exerting a significant amount of effort to complete this task. (b) The video shows a man driving a boat on a body of water. The boat is moving at a high speed, creating a large wake behind it. The man is wearing a green hat and appears to be enjoying the ride. The water is a deep blue color, and the sky is clear. The boat is white and appears to be a small motorboat. The man is the only person visible in the video.

## G    LIMITATIONS AND FUTURE WORK

Despite recent advances in multimodal large language models (MLLMs), their ability to reason and evaluate physical visual content remains limited. While these models perform well on tasks involving textual physical reasoning, they often struggle to interpret videos, particularly in evaluating the detailed alignment between video and text, or determining the physical commonsense of complex scenarios. Current outputs from these models tend to be shallow or generic, lacking the granularity required for tasks that demand nuanced feedback. This highlights the need for more robust mechanisms to process and reason over long-sequence video data.

To address these shortcomings, we propose two key directions. First, enhancing the quality and diversity of the physical video dataset is essential. Our current dataset comprises 8,000 curated videos, but scaling this further, *i.e.*, applying more fine-grained filtering and quality control, could significantly improve model learning. Second, future work should explore training strategies that explicitly target video-based reasoning and feedback generation, possibly through better failure cases solving mechanism. These steps could enable MLLMs to produce more detailed, context-aware assessments, ultimately closing the gap between textual and video understanding.

## H    THE USE OF LARGE LANGUAGE MODELS (LLMS)

We used an LLM to assist with writing, but all content in the paper was edited by the authors.

