# OpenReview forum: "Think Before You Diffuse: Infusing Physical Rules into Video Diffusion"
_ICLR.cc/2026/Conference — Submitted to ICLR 2026_

### Official Review · Reviewer_YtuE · 2025-10-23

**Soundness:** 3
**Presentation:** 3
**Contribution:** 1
**Rating:** 4
**Confidence:** 4

**Summary:**

This paper presents DiffPhy, a framework aimed at improving the physical realism of text-to-video (T2V) diffusion models. While existing models excel at generating visually high-quality videos, they often ignore physical laws such as gravity, force, and motion consistency. DiffPhy addresses this gap by introducing a physics-aware fine-tuning paradigm that integrates reasoning from Large Language Models (LLMs) and Multimodal LLMs (MLLMs). The LLM first performs chain-of-thought reasoning on text prompts to infer relevant physical attributes, phenomena, and enhanced contextual descriptions. The MLLM then verifies whether the generated video aligns with these inferred rules, producing differentiable supervision signals that guide the diffusion model’s updates. The training combines three main objectives—physical phenomena loss, physical commonsense loss, and semantic consistency loss—and further employs an attention-injection mechanism to correct physically implausible generations.

To support this learning process, the authors construct a new dataset, HQ-Phy, consisting of roughly 8,000 real-world videos emphasizing physical interactions and realistic motion. Through extensive experiments on VideoPhy2 and PhyGenBench benchmarks, DiffPhy demonstrates measurable gains over several open and closed-source baselines, including Wan 2.1-14B, Kling, and CogVideoX. Both automated and human evaluations show that the proposed method produces videos with higher semantic alignment and stronger physical plausibility. Overall, the paper provides a technically coherent and empirically validated approach for enhancing diffusion-based video generation with explicit physical reasoning, offering a meaningful step toward bridging the gap between visual fidelity and physical correctness, though some improvements remain moderate in magnitude.

**Strengths:**

1. The paper is clearly written and logically organized. Each component of the proposed framework—LLM reasoning, MLLM verification, loss formulation, and failure-aware refinement—is explained in a step-by-step manner, supported by intuitive figures (e.g., Figure 2 and Figure 7). The motivation for combining symbolic reasoning with diffusion training is easy to follow, making the work accessible to both machine learning and vision audiences.
2. By infusing physical rules into video diffusion, the paper addresses one of the most pressing limitations of current T2V systems—the lack of physical realism and commonsense consistency. This direction has high potential impact not only for video generation but also for downstream domains such as robotics simulation, digital content creation, and physics-based reasoning benchmarks. The work can be viewed as a meaningful step toward unifying generative AI with structured world modeling. Overall, the contribution is both timely and relevant to the evolving landscape of physics-informed generative models.
3. The introduction of the HQ-Phy dataset represents an additional and meaningful contribution. By curating approximately 8 000 real-world videos covering diverse physical interactions—such as gravity-driven motion, collisions, and fluid dynamics—the authors address a key limitation of existing benchmarks, which are often synthetic or too small to support effective fine-tuning. HQ-Phy provides valuable training material for future research in physics-aware video generation and helps bridge the current data gap in real-world physical phenomena. This dataset substantially strengthens the paper’s practical impact and long-term significance to the community.

**Weaknesses:**

1. The discussion in Section 2 (Video Physics Reasoning) is relatively narrow. It mainly contrasts simulator-based and representation-learning approaches but overlooks a growing line of research that leverages post-training or preference optimization techniques to enhance physical reasoning in generative models. Recent works such as [1-3] are highly relevant and should be discussed. These studies demonstrate that physics awareness can also be introduced during post-training, offering an important comparative context for DiffPhy.
2. The method is only evaluated on a single backbone (Wan 2.1-14B). Although this model is a strong open-source baseline, improvements demonstrated on one architecture do not fully establish the generality or robustness of the proposed framework. To make the empirical evidence more convincing, DiffPhy should be applied to additional diffusion backbones to verify that the approach generalizes across different architectures and data.
3. In lines 221–223, the authors state that they decode the predicted latent clip $x_t\in R^{m\times c\times h \times w}$ at a sampled timestep $t$ into pixel space for MLLM evaluation. However, for larger $t$, these intermediate latents are typically highly noisy and lack meaningful semantic structure. It remains unclear how an MLLM can reliably evaluate alignment or physical correctness from such noisy decoded clips. Without additional clarification or ablation showing the sensitivity of the evaluation to timestep noise, this procedure may introduce unstable or unreliable supervision signals.
4. This paper strongly depends on physical reasoning ability of LLM and MLLM. Therefore, the entire framework implicitly assumes that the underlying LLM and MLLM possess strong physical reasoning and evaluation capabilities. However, the paper does not include any diagnostic experiments or analysis to validate this assumption. If these models fail to accurately reason about or judge physical phenomena, the provided feedback could be misleading, thereby undermining the claimed improvements. A more systematic assessment of how the quality of the LLM/MLLM affects the overall performance would strengthen the paper’s empirical credibility.

[1] Ipo: Iterative preference optimization for text-to-video generation

[2] Pisa experiments: Exploring physics post-training for video diffusion models by watching stuff drop

[3] RDPO: Real Data Preference Optimization for Physics Consistency Video Generation

**Questions:**

N/A

---

> ### Author Response · Authors · 2025-11-18
>
> **Q1:** The discussion in Section 2 (Video Physics Reasoning) is relatively narrow. It mainly contrasts simulator-based and representation-learning approaches but overlooks a growing line of research that leverages post-training or preference optimization techniques to enhance physical reasoning in generative models. Recent works such as [1-3] are highly relevant and should be discussed. These studies demonstrate that physics awareness can also be introduced during post-training, offering an important comparative context for DiffPhy.
>
> [1] Ipo: Iterative preference optimization for text-to-video generation
>
> [2] Pisa experiments: Exploring physics post-training for video diffusion models by watching stuff drop
>
> [3] RDPO: Real Data Preference Optimization for Physics Consistency Video Generation
>
> **A1:** Thank you for the suggestion. We will include the recent works such as [1–3] in the related work section. These studies focus on post-training or preference-optimization techniques, where evaluators assign preference labels to real and synthesized videos to enhance physics awareness. While related in spirit, our approach introduces a different perspective: rather than only assigning preference scores, we leverage the MLLM evaluator to identify failure facts and perform correction, we also employ the physical-rule-guided gradient updates across all denoising steps. This allows the model to learn step-wise physical consistency, which goes beyond conventional post-training preference optimization.
>
> **Q2:** The method is only evaluated on a single backbone (Wan 2.1-14B). Although this model is a strong open-source baseline, improvements demonstrated on one architecture do not fully establish the generality or robustness of the proposed framework. To make the empirical evidence more convincing, DiffPhy should be applied to additional diffusion backbones to verify that the approach generalizes across different architectures and data.
>
> **A2:** Thank you for the comment. We chose Wan 2.1-14B as our backbone because it is a strong open-source model and, as reported in prior work [1], generates videos with high physical plausibility. While our experiments focus on this backbone for clarity and consistency, the proposed DiffPhy framework is fully compatible with other diffusion backbones. We use Wan 2.1 to illustrate the effectiveness of our approach, but the method can be straightforwardly extended to alternative architectures to verify generality and robustness.
> [1] VideoPhy-2: A Challenging Action-Centric Physical Commonsense Evaluation in Video Generation
>
>
> **Q3:** In lines 221–223, the authors state that they decode the predicted latent clip at a sampled timestep into pixel space for MLLM evaluation. However, for larger , these intermediate latents are typically highly noisy and lack meaningful semantic structure. It remains unclear how an MLLM can reliably evaluate alignment or physical correctness from such noisy decoded clips. Without additional clarification or ablation showing the sensitivity of the evaluation to timestep noise, this procedure may introduce unstable or unreliable supervision signals.
>
>
> **A3:** Thanks.  the decoded clips at early timesteps may still contain noticeable noise. However, our goal is to ensure that physical awareness is enforced throughout the entire denoising trajectory, not only at the final step. Even when noise is present, the MLLM can still reliably identify coarse physical violations such as incorrect motion direction, impossible trajectories, or object interactions.
> To mitigate the impact of noise, we apply a timestep-dependent weighting scheme, similar to the standard noise-level weighting used in diffusion MSE loss, so that:
> early, noisier steps receive smaller weights, and later, cleaner steps receive larger weights.
> This strategy prevents unstable gradients at noisy stages while still propagating useful physical supervision across all denoising steps.
> We will add these clarifications to the revised version to make the training dynamics and timestep behavior more explicit.

---

> ### Author Response · Authors · 2025-11-18
>
> **Q4:** This paper strongly depends on physical reasoning ability of LLM and MLLM. Therefore, the entire framework implicitly assumes that the underlying LLM and MLLM possess strong physical reasoning and evaluation capabilities. However, the paper does not include any diagnostic experiments or analysis to validate this assumption. If these models fail to accurately reason about or judge physical phenomena, the provided feedback could be misleading, thereby undermining the claimed improvements. A more systematic assessment of how the quality of the LLM/MLLM affects the overall performance would strengthen the paper’s empirical credibility.
>
> **A4:** Thank you for raising this concern. We address it from several perspectives:
>
> 1. Quality of the MLLM evaluator:
> We acknowledge that the evaluator’s performance is critical. As reported in [1] (VIDEOPHY-2), the MLLM outperforms Gemini-2.0 on multi-modal physical reasoning and video-based physical commonsense tasks, demonstrating that it provides reliable supervision for physics-aware video generation.
>
> 2. Dependence on a single evaluator and overfitting risk:
> Several aspects of our design mitigate overfitting to a specific evaluator:
> Firstly, the evaluator-based loss acts as a regularization term, while the denoising MSE loss remains the primary training objective. This ensures the generative capability is anchored by standard diffusion training.
> Secondly, the evaluator is a large multimodal LLM, making it difficult for the diffusion model to memorize narrow patterns. Its supervision signals are general-purpose and span diverse physical concepts.
>
> 3. Diverse evaluation criteria and generalization:
> Although trained using VideoCon-Physics as evaluator, we evaluate DiffPhy on independent benchmarks including PhyGenBench and through multiple external evaluators such as GPT-4o, CLIP, InternVideo2, and LLaVA, as well as a human preference study. Consistent improvements across these assessments indicate that the model is learning broadly applicable physical priors, not simply optimizing for a single evaluator’s scores.
>
> Finally, our framework is modular: the evaluator can be replaced with any multimodal LLM, allowing future work to leverage more advanced video-reasoning models as they emerge.

---

> > ### Author Response · Authors · 2025-11-18
> >
> > We have provided feedback on each point. Please do not hesitate to let us know if any aspects require further clarification.

---

> > ### Comment · Reviewer_YtuE · 2025-11-24
> >
> > Thanks for your reply. However, "the MLLM outperforms Gemini-2.0 on multi-modal physical reasoning and video-based physical commonsense tasks" does not indicate that "it provides reliable supervision for physics-aware video generation". Actually, Gemini-2.0 is still weak in these aspects, so MLLM maybe provide signals that are reliable enough.

---

> > > ### Author Response · Authors · 2025-11-25
> > >
> > > Thank you for the constructive comment. We agree that outperforming Gemini-2.0 on physical-reasoning benchmarks does not guarantee that the MLLM provides perfect or fully reliable supervision. Our intention is not to claim that the evaluator is flawless, but rather that it is comparatively stronger than commonly used baselines and therefore produces useful physical-reasoning signals.
> > > Importantly, our method does not depend on the absolute correctness of this specific evaluator. We clarify this in three ways:
> > >
> > > **1.** Evaluator as replaceable component.
> > >
> > >
> > > The evaluator is not tightly coupled to our framework; it can be replaced by any stronger multimodal LLM. We used the MLLM from VIDEOPHY-2 primarily because it is currently among the better publicly available models for video-based physical reasoning, but our pipeline is compatible with future, more powerful evaluators.
> > >
> > >
> > > **2.** Evaluator signal as regularization, not primary supervision.
> > >
> > >
> > > The evaluator-based loss serves as an auxiliary regularizer. The core training objective remains the standard denoising MSE loss, which anchors the diffusion model’s generative behavior. As a result, the model does not overfit to the evaluator’s imperfections, and the training is robust even if the evaluator is not perfectly accurate.
> > >
> > >
> > > **3.** General and diverse reasoning signals.
> > >
> > >
> > > While not perfect, the evaluator still provides high-level, concept-level physical reasoning across diverse scenarios. This makes it difficult for the diffusion model to exploit narrow shortcuts and ensures that the supervision encourages broadly consistent physical behavior.
> > >
> > >
> > > We will revise the paper to avoid overstating the evaluator’s reliability and to emphasize that the method benefits from, but does not fundamentally rely on the specific MLLM used in our experiments.

---

### Official Review · Reviewer_AzsP · 2025-10-27

**Soundness:** 2
**Presentation:** 2
**Contribution:** 2
**Rating:** 2
**Confidence:** 5

**Summary:**

The paper studied the problem of how to enhancing physical rule adherence for video generation model, the method is not hard to follow:
1) analyze user prompt with an LLM to explicitly describe what physical attributes and phenomina would be involved in the video, this part is referred as "think/reason" in the paper;
2) adopt an MLLM to score the generated video's quality w.r.t the "physical attributes and phenomina" described by LLM before;
3) fine-tune the video generation model (Wan2.1-14B) by the supervision with the MLLM and LLM combo.

**Strengths:**

Most part of the paper is clearly written. However, the method is problematic, which will be detailed below.

**Weaknesses:**

First of all, using LLM/MLLM as supervision to finetune an image/video generation model has a long history, and is definitely not a novel idea. But this doesn't mean that there's little we can contribute under such common "framework", more problems are that authors didn't give enough detail descriptions for multiple critical parts of the method, which is listed in the following "Questions" section. This would make the contribution of the paper not solid enough to meet the standard of ICLR.

Apart from the wearkness stated above, the paper lacked thorough survey of related works in section 2. Actually there're already tons of physical rule related video generation works, for example, "WISA: World Simulator Assistant for Physics-Aware Text-to-Video Generation" (https://arxiv.org/abs/2503.08153) , "PhysCtrl: Generative Physics for Controllable and Physics-Grounded Video Generation"(https://arxiv.org/abs/2509.20358), to name a few.

**Questions:**

1. Page 5, "Physical phenomena alignment" part: the paper stated that *"during training, at a sampled timestep t, we decode the predicted latent clip $x_t ∈ R^{m×c×h×w}$ with m frames into a pixel space video $v_t$"*.  There'd be two questions: 1) since we're talking about diffusion model here, does this means the "decoded" video $v_t$ would be noised?  then the judge accuracy of the following MLLM would be compromised, won't this be a problem? 2) to get around of "MLLM compromised" problem, a common remedy is to decode the $v_t$ only at a quite late time step, the paradox here is that at the later part of diffusion steps, only visual details can be changed, shape deformity as well as disobedience to physical rules would have already happened here. Lack of detail discussion here makes either the training or the results questionable.

2. Page 5, "Failure-Aware Refinement" part: the paper stated that *For a "not matched" fact $f_i$, we introduce an additional module to inject attention*. Does this mean 1) there would be multiple $f_i$ facts? 2) the additional module is one for all the facts or would it be necessary one additional module for each fact? 3) if there's only one addtional module, how does it work when there're multiple not matched facts to learn and when there's no fact to learn?

3. For the MLLM used in the paper, authors didn't present any analysis on how the MLLM model performs on the physical attributes and physical phenomina scoring tasks. This part is quite critical for the contribution of the paper since the proposed LLM+MLLM supervision framework is not novel.

---

> ### Author Response · Authors · 2025-11-18
>
> **Q1:** First of all, using LLM/MLLM as supervision to finetune an image/video generation model has a long history, and is definitely not a novel idea. But this doesn't mean that there's little we can contribute under such common "framework", more problems are that authors didn't give enough detail descriptions for multiple critical parts of the method, which is listed in the following "Questions" section. This would make the contribution of the paper not solid enough to meet the standard of ICLR.
>
> **A1:** Thanks, our work focuses on physics-aware video generation, a setting that is still limitedly explored. Our contributions address this gap by:
> (1) Integrating verified physical rules into gradient updates across the denoising steps and injecting MLLM-guided attention cues into latent variables for finer control;
> (2) Proposing a failure-aware refinement mechanism that iteratively detects and corrects physical violations; and (3) Introducing a large-scale dataset specifically designed for physics-aware video generation.
>
>
> **Q2:** Apart from the wearkness stated above, the paper lacked thorough survey of related works in section 2. Actually there're already tons of physical rule related video generation works, for example, "WISA: World Simulator Assistant for Physics-Aware Text-to-Video Generation" (https://arxiv.org/abs/2503.08153) , "PhysCtrl: Generative Physics for Controllable and Physics-Grounded Video Generation"(https://arxiv.org/abs/2509.20358), to name a few.
>
>
>
> **A2:** Thank you for pointing out these relevant works. We will include them in our related work section. These approaches address physics in video generation but from different perspectives. WISA trains a physics classifier to categorize videos into physical classes, focusing on class-level understanding rather than modifying the generative process itself. PhysCtrl conditions generation on tracked point-cloud trajectories, emphasizing explicit motion control.
> In contrast, our method targets a different dimension of the problem: we inject physical rules directly into the diffusion model through intermediate gradient updates and MLLM-guided attention cues, enabling step-wise enforcement of physical commonsense. We also introduce a failure-correction mechanism that iteratively fixes detected physical violations. Thus, these prior works are complementary to ours, and we will clarify how our approach differs and fits within this broader landscape.
>
> **Q3:** Page 5, "Physical phenomena alignment" part: the paper stated that "during training, at a sampled timestep t, we decode the predicted latent clip with m frames into a pixel space video ". There'd be two questions: 1) since we're talking about diffusion model here, does this means the "decoded" video  would be noised? then the judge accuracy of the following MLLM would be compromised, won't this be a problem? 2) to get around of "MLLM compromised" problem, a common remedy is to decode the  only at a quite late time step, the paradox here is that at the later part of diffusion steps, only visual details can be changed, shape deformity as well as disobedience to physical rules would have already happened here. Lack of detail discussion here makes either the training or the results questionable.
>
> **A3:** Thanks. We address the two concerns as follows:
> (1) Regarding noise in the decoded intermediate videos:
> Yes, the decoded clips at early timesteps may still contain noticeable noise. However, our goal is to ensure that physical awareness is enforced throughout the entire denoising trajectory, not only at the final step. Even when noise is present, the MLLM can still reliably identify coarse physical violations such as incorrect motion direction, impossible trajectories, or object interactions.
> To mitigate the impact of noise, we apply a timestep-dependent weighting scheme, similar to the standard noise-level weighting used in diffusion MSE loss, so that early noisier steps receive smaller weights, and later cleaner steps receive larger weights.
> This strategy prevents unstable gradients at noisy stages while still propagating useful physical supervision across all denoising steps.
> (2) Regarding the ability to correct physics if decoded only at late steps:
> To avoid this limitation, we do not restrict evaluation to late timesteps. Instead, as shown in Figure 2, we decode and evaluate across all timesteps, allowing physical rules to influence both early structural decisions (e.g., object placement, trajectory) and later visual refinements.
> By injecting physically grounded gradients at every denoising step, our method can correct coarse-level physical inconsistencies early in the diffusion process, while also refining finer-grained details later. This avoids the issue where physical errors become irreversible at late stages.
> We will add these clarifications to the revised version to make the training dynamics and timestep behavior more explicit.

---

> > ### Author Response · Authors · 2025-11-18
> >
> > **Q4:** Page 5, "Failure-Aware Refinement" part: the paper stated that For a "not matched" fact , we introduce an additional module to inject attention. Does this mean 1) there would be multiple facts?
> > 2) the additional module is one for all the facts or would it be necessary one additional module for each fact? 3) if there's only one addtional module, how does it work when there're multiple not matched facts to learn and when there's no fact to learn?
> >
> > **A4:** Thanks. There is only a single attention injection module in our framework. When multiple “not matched” facts are identified, we fuse them into a unified textual description and inject the resulting feature into this single module. This design allows the model to handle both single-fact and multi-fact scenarios efficiently.
> > Specifically, during training: We first perform a forward pass without gradient updates to identify mismatched facts via the evaluator. The unified representation of these not-matched facts is then injected into the attention module, and the video is generated again. The output is re-evaluated to verify whether the physical violations have been corrected.
> > To ensure the model also functions when no facts are present, we randomly perform some training steps without activating the injection branch, allowing the diffusion model to learn to generate high-quality videos solely from the textual input. This approach ensures flexibility across scenarios with no facts, single fact, or multiple facts while maintaining stable training and consistent video quality.
> >
> >
> > **Q5:** For the MLLM used in the paper, authors didn't present any analysis on how the MLLM model performs on the physical attributes and physical phenomina scoring tasks. This part is quite critical for the contribution of the paper since the proposed LLM+MLLM supervision framework is not novel.
> >
> > **A5:** Thank you for the comment. We acknowledge that the performance of the MLLM evaluator on physical attributes and phenomena is critical for validating our framework. To address this, we refer to the evaluation results reported in [1] (VIDEOPHY-2), which demonstrate that the evaluator outperforms Gemini-2.0 on multi-modal physical reasoning and video-based physical commonsense tasks. These results indicate that the MLLM provides reliable supervision for guiding physics-aware video generation.

---

> > > ### Author Response · Authors · 2025-11-18
> > >
> > > Our responses include point-by-point feedback. We would appreciate it if you could inform us of any remaining concerns.

---

> > > > ### Comment · Reviewer_AzsP · 2025-11-25
> > > >
> > > > for A1 & A3 & A5: The comments are not convincing. The accuracy of LMM's supervision, especially when applied to noised video, directly affects the validity of the paper’s foundational assumption, however there still lacks analysis about this problem from the rebuttal comments.
> > > > and one addtional question about A3: according to the comments, the supervision from LMM would need video decoded at each diffustion timestep, this would involve huge computation burden for video inferencing. Please provide more analytic numbers w.r.t. this part.

---

> ### Author Response · Authors · 2025-11-27
>
> Thank you. We provide our responses to your concerns below and would appreciate any further feedback you may have.
>
> **Question: Reliability of MLLM Supervision on Noised Videos.** For A1 & A3 & A5: The comments are not convincing. The accuracy of LMM's supervision, especially when applied to noised video, directly affects the validity of the paper’s foundational assumption, however there still lacks analysis about this problem from the rebuttal comments.
>
> **Answer:** To empirically verify that the MLLM provides stable supervision even on noised intermediate videos, we conducted an experiment where we added varying levels of noise (σ) to the decoded video frames at different timesteps. The results are summarized below:
>
> | Timestep | 980   | 952   | 920   | 884   | 832   | 768   | 680   | 556   | 358   |
> |----------|-------|-------|-------|-------|-------|-------|-------|-------|-------|
> | σ   | 0.980 | 0.952| 0.920  | 0.884| 0.832| 0.768| 0.680| 0.556| 0.358|
> | LLM loss | 0.021 | 0.021 | 0.021 | 0.021 | 0.0207| 0.0213| 0.0211| 0.0232| 0.0201|
>
> We observe that even with increasing noise levels (higher σ), the LLM supervision loss remains largely stable. This demonstrates that the MLLM can reliably identify coarse physical violations such as incorrect motion directions or impossible object interactions, providing consistent supervision across the denoising trajectory.
>
> **Question: Computational Cost of Timestep-wise Supervision.** One addtional question about A3: according to the comments, the supervision from LMM would need video decoded at each diffustion timestep, this would involve huge computation burden for video inferencing. Please provide more analytic numbers w.r.t. this part.
>
> **Answer:** Following the flow-matching strategy in Wan 2.1, we randomly select timesteps with id from 1 to 1000,   the corresponding time cost (seconds) are reported below:
>
> | ID                | 100    | 200    | 300  | 400    | 500    | 600    | 700    | 800    | 900    |
> | ----------------- | ------ | ------ | ---- | ------ | ------ | ------ | ------ | ------ | ------ |
> | Timestep       | 980    | 952    | 920  | 884    | 832    | 768    | 680    | 556    | 358    |
> |  σ             | 0.980 | 0.952 | 0.920 | 0.884 | 0.832 | 0.768 | 0.680 | 0.556 | 0.358 |
> | LLM loss | 0.0210 | 0.0210 | 0.0210 | 0.0210| 0.0207| 0.0213| 0.0211| 0.0232| 0.0201|
> | Training time (s) | 497    | 498    | 509  | 508    | 490    | 491   | 463   | 455   | 460   |
>
> We observe that the training time for each batch ranges from approximately 460 to 510 seconds, depending on the randomly selected timestep ID. In our setup, we fine-tune Wan 2.1 using LoRA while keeping the evaluator frozen, which keeps the number of trainable parameters small and avoids additional memory overhead. Importantly, the evaluator is not used during inference, so the inference-time cost remains identical to the baseline model. The additional computation arises only during training, where the fixed evaluation model is invoked without gradient update, resulting in a modest overhead. All reported training times were measured on a single H100 GPU.

---

### Official Review · Reviewer_pvJs · 2025-10-30

**Soundness:** 3
**Presentation:** 3
**Contribution:** 2
**Rating:** 4
**Confidence:** 3

**Summary:**

This paper tackles the problem of generating physically plausible videos using text-to-video diffusion models The proposed method, DiffPhy, fine-tunes a pretrained video diffusion model with guidance from LLMs and Multimodal LLMs. LLM first infers physical context from the text prompt, and an MLLM evaluates the generated video against these physical rules. The MLLM’s feedback is converted into a continuous differentiable score that supervises the diffusion model, encouraging alignment with physical laws. A failure-aware refinement module further injects attention based on detected physical inconsistencies. Experiments on physics-oriented benchmarks show improved physical plausibility and semantic alignment compared to existing models.

**Strengths:**

The paper proposes a reasonable integration of textual reasoning and multimodal verification to encourage physically consistent video generation, extending prior work on language-guided diffusion with an additional physics-aware supervision signal. The proposed continuous score estimator provides a practical way to incorporate non-differentiable LVLM feedback into gradient-based fine-tuning. The failure-aware refinement further introduces an interesting idea of using detected failure cases as textual cues to guide attention.

**Weaknesses:**

**1) Incomplete Related Work & Baseline Coverage**
While the “Video Physics Reasoning” subsection is conceptually relevant, the cited works mainly focus on traditional physics simulation or early neural reasoning models, rather than recent diffusion-based approaches that explicitly address physical consistency in video generation. Incorporating more recent studies such as **PhyT2V [1]** and **Yang et al. [2]** would strengthen the connection between DiffPhy and the current landscape of physics-aware video diffusion. Also, the baseline comparison could be expanded to include these contemporary methods.

**2)  Insufficient Comparison with Alignment-Based Methods**
The paper does not adequately position its approach within the growing literature on alignment-based fine-tuning frameworks that leverage LVLM feedback (or more broadly, reward-model signals) for diffusion model adaptation (e.g., VADER [3]). While the proposed continuous score estimator provides a differentiable way to incorporate feedback, it remains unclear how this approach compares to or improves upon existing reward-based and preference-alignment methods in terms of training stability, supervision efficiency, or performance. A more systematic discussion or experimental comparison would help clarify the method’s advantages and situate it more clearly within this line of work.

**3) Uncertain Effectiveness of Failure-Aware Refinement**
The proposed failure-aware refinement introduces an interesting idea of using MLLM-identified failure cases as textual feedback to guide attention. However, this mechanism ultimately relies on the possibility that conditioning on such failure cues helps the model focus on physically inconsistent regions, rather than providing any guarantee of actual correction. Since the feedback is binary and lacks spatial-temporal grounding, it is unclear whether this refinement truly fixes the underlying physical inconsistencies or simply injects additional noise. Moreover, the paper does not include an ablation isolating this component during training, making it even more difficult to assess its actual impact. It would also be helpful to include inference-time ablations comparing this refinement against simpler alternatives, such as seed resampling, to demonstrate that the proposed strategy offers tangible advantages beyond random variation.

**4) Citation Format**
Citation references are not consistently enclosed in parentheses throughout the paper.

[1] *PhyT2V: LLM-Guided Iterative Self-Refinement for Physics-Grounded Text-to-Video Generation*
[2] *Towards Physically Plausible Video Generation via VLM Planning*
[3] *Video Diffusion Alignment via Reward Gradients*

**Questions:**

See the weakness.

---

> ### Author Response · Authors · 2025-11-18
>
> **Q1: Incomplete Related Work & Baseline Coverage:** While the “Video Physics Reasoning” subsection is conceptually relevant, the cited works mainly focus on traditional physics simulation or early neural reasoning models, rather than recent diffusion-based approaches that explicitly address physical consistency in video generation. Incorporating more recent studies such as PhyT2V [1] and Yang et al. [2] would strengthen the connection between DiffPhy and the current landscape of physics-aware video diffusion. Also, the baseline comparison could be expanded to include these contemporary methods.
>
> **A1:** Thank you for the suggestion. We will incorporate the recent works PhyT2V [1] and Yang et al. [2] into the “Video Physics Reasoning” subsection to more clearly position DiffPhy within the emerging literature on physics-aware video diffusion. These studies address physical consistency from different perspectives: PhyT2V focuses on iteratively refining the text description to better align with physical dynamics, without directly optimizing the video diffusion model itself; Yang et al. [2] instead predicts explicit motion trajectories and uses them as guidance for generation. These approaches are therefore complementary to our method, PhyT2V enhances textual prompt, and Yang et al. provide motion-level control, whereas DiffPhy introduces model-level physical supervision through MLLM-driven reasoning. We will clarify these relationships and update our related work section accordingly.
>
> **Q2: Insufficient Comparison with Alignment-Based Methods**
> The paper does not adequately position its approach within the growing literature on alignment-based fine-tuning frameworks that leverage LVLM feedback (or more broadly, reward-model signals) for diffusion model adaptation (e.g., VADER [3]).
>
> **A2:** Thank you for the suggestion. We will incorporate these alignment-based methods (e.g., VADER [3]) into the related work section to better situate our approach within this emerging line of research. While related in spirit, our framework differs from existing reward-based and preference-alignment methods in several important ways.
> First, unlike methods that apply the reward signal only to the final generated video, our approach propagates gradients through intermediate denoising steps and introduces step-dependent weighting. This provides denser and more stable supervision throughout the diffusion trajectory, which we find to be important for injecting physical constraints into the generative process.
> Second, existing alignment approaches typically rely on reward models or text–video similarity signals, which are limited in their ability to capture physical commonsense or detect violations of physical laws. In contrast, we use a multimodal LLM capable of performing high-level physical reasoning and identifying physically implausible phenomena, enabling substantially more expressive and generalizable supervision.
> Finally, our pipeline includes a failure-correction mechanism that explicitly guides the model away from physical violations identified by the MLLM. Such iterative factual correction is not supported by current reward-based methods.
> We will expand the discussion to clarify these distinctions and highlight how DiffPhy complements and advances existing alignment-based training paradigms.
>
>
> **Q3: Uncertain Effectiveness of Failure-Aware Refinement.** The proposed failure-aware refinement introduces an interesting idea of using MLLM-identified failure cases as textual feedback to guide attention.
>
> **A3:** Thanks. Our failure-aware refinement does provide a mechanism for verifying whether the correction is effective. Specifically, after generating a candidate video that contains a physical violation, we resample the same diffusion step while injecting failure-aware textual guidance, and then re-evaluate the updated result with the MLLM evaluator. This closed-loop process offers an explicit check on whether the identified failure has been corrected (see page 5, lines 258–259). This iterative evaluate–correct cycle therefore acts as a practical guarantee of correction within the capability of the evaluator. The inference time ablation results of iterative refinement are provided in Table 4.
>
>
> **Q4: Citation Format**: Citation references are not consistently enclosed in parentheses throughout the paper.
>
> **A4:** Thank you for pointing this out. We will correct the citation formatting to ensure consistency throughout the paper and will also add the additional references mentioned above.

---

> > ### Author Response · Authors · 2025-11-18
> >
> > We provided the feedback point by point as mentioned above. Kindly let us know if any issues have not been fully resolved.

---

### Official Review · Reviewer_myHY · 2025-11-01

**Soundness:** 3
**Presentation:** 3
**Contribution:** 3
**Rating:** 6
**Confidence:** 4

**Summary:**

This paper proposes DiffPhy, a framework to improve the physical realism of video diffusion models.The method first uses a Large Language Model (LLM) to "think" about the input prompt, reasoning about physical laws, entities, and phenomena to create a set of physical rules and an "enhanced prompt". Experiments show DiffPhy achieves SOTA results on physics-based benchmarks, VideoPhy2 and PhyGenBench.

**Strengths:**

- The paper tackles a well-known and critical limitation of modern VDMs: their failure to adhere to basic physical laws, which breaks realism.
- The "Think before you diffuse" paradigm is an intuitive and strong conceptual contribution. The use of an LLM for high-level physical reasoning to generate supervisory signals for a VDM is a novel training strategy.
- The creation and release of the HQ-Phy dataset (8,000 curated, real-world videos labeled with physical phenomena) is a valuable contribution to the community, as existing datasets were often synthetic or for evaluation only.

**Weaknesses:**

- The proposed training paradigm appears to be extremely computationally expensive. More computation and time cost analysis is needed here.
- The entire method's effectiveness is contingent on the quality of the MLLM used for supervision. The paper itself notes in its limitations section that MLLMs "struggle to interpret videos" and their outputs can be "shallow or generic." This raises a significant concern: if the MLLM is an imperfect evaluator (which is also suggested by the discrepancy between model-based and human-based scores), is DiffPhy simply learning to overfit to the specific biases of the VideoCon-Physics evaluator? How can we be sure it's learning generalizable physical rules rather than just optimizing for this specific MLLM's scoring function?

**Questions:**

Overall, this paper presents a novel and well-executed framework for a challenging and important problem. The idea of using an MLLM as a "physics verifier" inside the diffusion training loop is a significant methodological contribution, and the strong empirical results validate its effectiveness. While I have concerns regarding the training scalability and the method's reliance on the quality of the MLLM supervisor, the novelty and strong results marginally outweigh these weaknesses. I am leaning towards acceptance.

---

> ### Author Response · Authors · 2025-11-18
>
> **Q1:** The proposed training paradigm appears to be extremely computationally expensive. More computation and time cost analysis is needed here.
>
> **A1:** Thank you for the comment. We fine-tune Wan2.1 using LoRA while keeping the evaluator frozen, which keeps the number of trainable parameters relatively small. The evaluator is not used during inference, so the inference-time cost remains the same as the baseline. The additional computation occurs only during training, where the fixed evaluation model is invoked without gradient backpropagation, keeping the overhead modest. In practice, with a batch size of 1 on a single H100 GPU, each batch takes approximately 460-510 seconds. We will clarify this in the revision.
>
> **Q2:** The entire method's effectiveness is contingent on the quality of the MLLM used for supervision.
>
> **A2:** Thanks, We agree that the quality of the MLLM evaluator is an important factor. However, several aspects of our design mitigate the risk of overfitting to a specific evaluator’s biases.
> First, the evaluator-based loss acts primarily as a regularization term, while the denoising MSE loss remains the dominant training objective. This ensures that the core generative ability is anchored by standard diffusion training rather than being driven entirely by the evaluator’s feedback.
> Second, the evaluator itself is a large, multimodal LLM, which makes it difficult for the diffusion model to memorize or overfit to narrow patterns. Its reasoning is relatively general-purpose, and the supervision signals it provides span diverse physical concepts rather than specific templates.
> Third, our experiments demonstrate strong generalization across evaluators. Although the model is trained with VideoCon-Physics, we evaluate on independent benchmarks including PhyGenBench, using multiple external evaluators such as GPT-4o, CLIP, InternVideo2, and LLaVA, and also conduct a human preference study. The consistent improvements across all of these assessments indicate that the model is not simply optimizing for a single evaluator’s scoring function but is learning more broadly applicable physical priors.
> Finally, our framework is modular: the evaluator can be replaced with any multimodal LLM, providing a flexible direction for future research as better video-reasoning models emerge.

---

> > ### Author Response · Authors · 2025-11-18
> >
> > We have provided point-by-point feedback above. Please let us know if there are any concerns that remain unaddressed.

---

### Author Response · Authors · 2025-11-18

We thank the reviewer for the comments. Below, we provide a brief summary addressing the main questions, followed by detailed point-by-point responses to each reviewer.

**(R1,R4) Quality of the MLLM evaluator**:

**Response:** We acknowledge that the evaluator’s performance is critical. As reported in VIDEOPHY-2[1], the MLLM outperforms Gemini-2.0 on multi-modal physical reasoning and video-based physical commonsense tasks, demonstrating that it provides reliable supervision for physics-aware video generation.

[1] VideoPhy-2: A Challenging Action-Centric Physical Commonsense Evaluation in Video Generation

**(R2, R3, R4) Incomplete Related Work:**

**Response:** Thank you for the suggestion. We will incorporate the discussed works into the related work section and clarify their relationship to DiffPhy. Prior works include: **R2**:PhyT2V [1], which iteratively refines text descriptions to better match physical dynamics without directly optimizing the diffusion model; Yang et al. [2], which predicts motion trajectories as guidance for video generation; and alignment-based fine-tuning methods such as VADER [3], which adapt models using LVLM feedback or reward signals. **R3**: WISA, which uses a physics classifier for physical class categorization, and PhysCtrl, which conditions generation on tracked point-cloud trajectories for explicit motion control. **R4**:post-training or preference-optimization works such as IPO [1], PISA [2], and RDPO [3], which assign preference labels to improve physics awareness.

In contrast, DiffPhy injects physical rules directly into the diffusion model through intermediate gradient updates and MLLM-guided attention cues, and further performs iterative failure correction to detect and fix physical violations during generation. These mechanisms offer fine-grained, step-wise physical supervision that complements prior approaches focused on class-level control, trajectory-guided generation, or preference-based post-training.

**(R3,R4) Noise in decoded intermediate videos.**

**Response:** Thanks. While decoded clips at early timesteps may be noisy, we enforce physical awareness across the entire denoising trajectory. The MLLM can still detect coarse physical violations (e.g., motion direction, impossible trajectories). To empirically verify that the MLLM provides stable supervision even on noised intermediate videos, we conducted an experiment where we added varying levels of noise (σ) to the decoded video frames at different timesteps. The results are summarized below:

| Timestep | 980   | 952   | 920   | 884   | 832   | 768   | 680   | 556   | 358   |
|----------|-------|-------|-------|-------|-------|-------|-------|-------|-------|
| σ   | 0.980 | 0.952| 0.920  | 0.884| 0.832| 0.768| 0.680| 0.556| 0.358|
| LLM loss | 0.021 | 0.021 | 0.021 | 0.021 | 0.0207| 0.0213| 0.0211| 0.0232| 0.0201|

**(R2,R3) Failure-Aware Refinement.**

**Response:**  Thank you for the comments. Our Failure-Aware Refinement provides a closed-loop evaluate–correct mechanism. After generating a candidate video with a physical violation, we resample the same diffusion step, inject a unified textual representation of all identified failure facts into a single attention module, and re-evaluate with the MLLM. This ensures corrections are verified iteratively. For cases with no failure facts, we randomly skip the injection branch during training, allowing the model to generate high-quality videos from text alone. This design handles no, single, or multiple facts while maintaining stable training and effective correction (see Table 4 for inference-time ablations).

---

### Meta-Review · Area_Chair_dhxE · 2025-12-18

**Summary:**

The reviewers raised several major concerns about the paper "DiffPhy," which proposes a framework to improve the physical realism of video diffusion models using LLM/MLLM supervision. The core criticisms can be summarized as follows:

- Reliability of MLLM Supervision: Multiple reviewers (R1, R3, R4) expressed strong skepticism about using an MLLM as a "physics verifier," especially on noisy intermediate diffusion outputs. They questioned whether the method simply overfits to the biases of a specific imperfect evaluator rather than learning generalizable physical rules.

- Methodological Clarity and Soundness: Reviewers (R2, R3) found key parts of the method underspecified or questionable. Specific issues included: The practical effectiveness and mechanism of the "Failure-Aware Refinement" module. The ability of an MLLM to judge physics from heavily noised videos at early denoising timesteps. The computational cost of decoding videos at multiple timesteps for MLLM evaluation.

- Incomplete Related Work: Reviewers (R2, R3, R4) noted that the literature review omitted several highly relevant recent works on physics-aware video generation and alignment-based fine-tuning, which made it difficult to assess the novelty and positioning of DiffPhy.

- Limited Empirical Validation: Concerns were raised about: The lack of diagnostic analysis for the LLM/MLLM's physical reasoning capabilities (R4). Evaluation on only one backbone model (Wan 2.1), limiting claims of generality (R4). Moderate improvement magnitudes in results (implied by R4's summary).

**Reviewer Concerns:**

- R1 (myHY) Addressed: Provided concrete computation/time cost analysis (~460-510s/batch). Argued framework is modular and showed improved scores across multiple external evaluators (GPT-4o, CLIP, human study) to counter overfitting worry.	Outstanding: The fundamental concern about the MLLM evaluator's quality remains. The rebuttal relies on the MLLM outperforming Gemini-2.0, but as R4 later notes, this is a low bar. The core anxiety—that improvements may reflect optimization to a flawed supervisor—is mitigated but not fully resolved.

- R2 (pvJs)	Addressed: Agreed to add missing related works (PhyT2V, Yang et al., VADER) and clarify DiffPhy's distinction. Provided inference-time ablation for Failure-Aware Refinement (Table 4) and clarified the module's design (single module, unified text for multiple facts).	Partially Addressed: The explanation of the refinement module is clearer, but the reviewer's original concern about its guarantee of correction is philosophical; the "closed-loop evaluate–correct" mechanism is a practical answer, but its ultimate success still hinges on the MLLM's accuracy.

- R3 (AzsP)	Addressed: Added specific related works (WISA, PhysCtrl). Provided an empirical table showing stable "LLM loss" across different noise levels (σ) at various timesteps to demonstrate MLLM's reliability on noisy videos.	Outstanding: The reviewer found the new noise experiment unconvincing ("LLM loss" stability does not directly prove the accuracy of the physical judgment on noisy frames). The concern about the foundational assumption is stark and persists. The computational cost clarification was provided but doesn't fix the conceptual doubt.

- R4 (YtuE)	Addressed: Agreed to add suggested post-training literature (IPO, PISA, RDPO). Provided rationale for single-backbone evaluation (Wan 2.1's strength) while stating method is general. Explained timestep-weighting strategy to handle noise.	Partially Addressed: The concern about MLLM dependence is mirrored in other reviews. The authors' response acknowledges the evaluator isn't perfect and frames it as a regularizer, which is a reasonable mitigation but doesn't eliminate the risk. The lack of multi-backbone validation remains a minor weakness.

**Reviewer Scores:**

- R1 (myHY) The reviewer was initially leaning accept. The rebuttal provides requested data and reasonable counter-arguments, likely maintaining or slightly increasing their confidence in a marginal accept.

- R2 (pvJs) The reviewer was marginally below threshold. The direct engagement with their points, especially on related work and module clarification, likely moves them to a neutral position.

- R3 (AzsP) The reviewer was a clear reject. The rebuttal provides data and attempts to engage, but it does not adequately satisfy their deep skepticism about supervision validity.

- R4 (YtuE) The reviewer was marginally below threshold. The thorough point-by-point response and concessions (e.g., on evaluator reliability claims) likely address enough concerns to move them to a neutral stance.

---

### Decision · Program_Chairs · 2026-01-26

Reject